# MuseCoco: Generating Symbolic Music from Text

## Abstract

Due to the inherent ease of textual input for user engagement, it is natural to generate music from text. In this paper, we introduce MuseCoco (Music **Co**mposition **Co**pilot), a system meticulously designed to compose symbolic music from text descriptions. It operates by utilizing musical attributes as a bridge and dividing the process into text-to-attribute understanding stage and attribute-to-music generation stage, which bestows three key advantages: First, it removes the necessity for paired text-to-music data by automatically generating text-attribute pairs for the text-to-attribute understanding stage and extracting attributes directly from music data for the attribute-to-music generation stage. This alleviates the labor-intensive process of human annotation. Second, with its clear attribute design, the system provides precise control over musical elements, ensuring accurate shaping of the musical output according to the user's intentions. Third, it enhances versatility and usability by providing an additional option for attribute-conditioned control beyond textual input. Our experimental results demonstrate that MuseCoco significantly outperforms our top-performing baseline model, GPT-4, on musicality, controllability, and overall score, by 45.5%, 35.2%, and 47.0%, respectively. There is also a notable enhancement of approximately 20% in objective control accuracy. Additionally, we have developed a large-scale model with 1.2 billion parameters, showcasing exceptional controllability and musicality. In practical applications, MuseCoco can serve as a user-friendly tool for musicians, enabling them to effortlessly generate music by simply providing text descriptions, and offering a substantial enhancement in efficiency compared to manually composing music from scratch. Music samples generated by MuseCoco are available via this link [1].

## 1 Introduction

Text-to-music generation stands as a pivotal task in the realm of automatic music generation, as it can empower users to craft music effortlessly and intuitively by employing natural language as a creative interface. It renders music generation remarkably user-friendly, especially for individuals lacking a foundational background in music theory or composition.

Currently, there exist models capable of generating musical audio from texts (Agostinelli et al., 2023; Huang et al., 2023; Zhu et al., 2023; Schneider et al., 2023), and they have achieved considerable success. However, they also exhibit certain drawbacks mainly due to intrinsic properties of audio: 1) Limited editability: Musical audio is represented as waveforms, which is hard to explicitly encode musical elements like music notes and instrument details. Consequently, once the audio is produced, making modifications to the musical elements becomes challenging unless restarting the generation process. 2) Lack of control: Generating musical audio from textual descriptions often lacks precise control over specific musical attributes such as tempo, meter, and rhythm. This limitation arises because the generation process lacks explicit control mechanisms.

In contrast to audio, symbolic music, a form of musical notation that employs symbols to convey specific musical ideas, is well-suited to overcome the aforementioned drawbacks and align with users' demands for editing and precise control. Several works (Zhang et al., 2020; Wu & Sun, 2022; OpenAI, 2023) have demonstrated the capability to generate symbolic music from text. However,

---

[1] https://musecoco.github.io

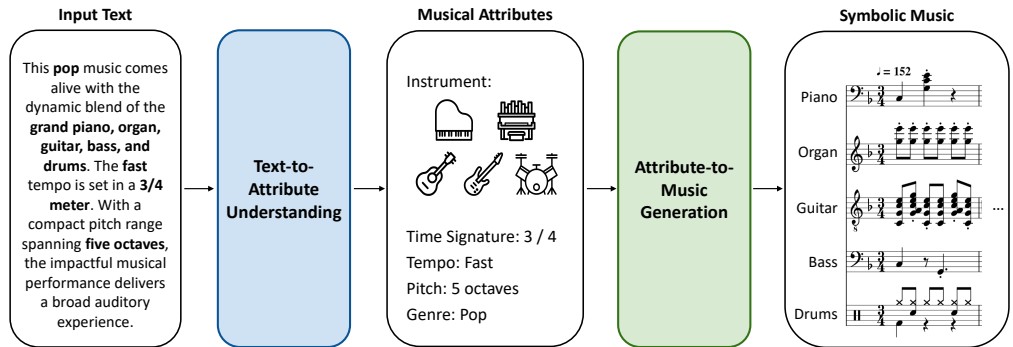

Figure 1: The two-stage framework of MuseCoco. Text-to-attribute understanding extracts diverse musical attributes, based on which symbolic music is generated through the attribute-to-music generation stage.

they still contend with issues such as unnatural text descriptions and subpar performance. Specifically, BUTTER (Zhang et al., 2020) is constrained by specific textual inputs, impeding its ability to generalize to a more natural textual input. Furthermore, it can only exert control over limited music aspects, restricting the model's capacity to capture the full spectrum of musical creativity and failing to meet many user requirements. As for the BART-based music generation model (Wu & Sun, 2022) and GPT-4 (OpenAI, 2023), although they allow natural language inputs, their control accuracy and musical quality may fall short of expectations, mainly due to the scarcity of extensive paired text-music data upon which they heavily rely. In addition, although Mubert[1] possesses the ability to produce editable MIDI compositions, it primarily combines pre-existing music pieces instead of generating novel ideas, limiting its responsiveness to input prompts.

To solve the aforementioned drawbacks and issues, we propose MuseCoco (Music **Co**mposition **Co**pilot), a two-stage system for generating symbolic music from text descriptions. It leverages musical attributes as the bridge, and breaks down the text-to-music generation into two stages: text-to-attribute understanding and attribute-to-music generation, as shown in Figure 1. In text-to-attribute understanding, the values of a set of musical attributes (such as instrument, rhythm, tempo, detailed in Table 1) are extracted from the textual input. In attribute-to-music generation, it generates symbolic music that conforms to the music attributes.

The proposed two-stage design brings substantial benefits to this task. First, it is data efficient. In the attribute-to-music generation stage, musical attributes can be easily extracted from music sequences or obtained from existing attribute-labeled datasets, allowing the model in the attribute-to-music generation stage to be trained in a self-supervised manner. In the text-to-music understanding stage, we synthesize paired text-to-attribute data by creating text templates for describing each attribute, combining a subset of these templates, and further refining them into a coherent paragraph using ChatGPT's language generation capabilities. Without the need of paired text-music data, a large amount of symbolic music data can be leveraged, helping to improve model performance by increasing data amount and model size simultaneously. Second, it can achieve a more precise control. Using attributes as conditions can help explicit control in generating music across multiple aspects, enabling fine-grained manipulation and customization of the generated musical output. Third, it provides multiple ways of controlling. Users have the option to directly describe the music they wish to generate using textual inputs, which stands as the primary and advantageous method. We also offer an alternative option for advanced musicians well-versed in music theory, allowing them to input attribute values directly into the second stage to create compositions, providing an extra layer of versatility.

The main contributions of this work are as follows:

- We introduce MuseCoco, a system that seamlessly transforms textual input into musically coherent symbolic compositions. This innovative approach empowers musicians and general users from diverse backgrounds to create music more efficiently and with better control.

---
[1] https://mubert.com/

- With this two-stage framework, a large amount of symbolic data can be used without the need for labeled text descriptions. It offers users two engagement options: leveraging text descriptions or directly specifying attribute values to control the music generation process.

- Subjective evaluations demonstrate that MuseCoco surpasses baseline systems across musicality, controllability, and overall performance, with minimum improvements of 45.5%, 35.2%, and 47.0%, respectively. Moreover, there is a noteworthy enhancement of approximately 20% in objective control accuracy. Our model's scalability to 1.2 billion parameters further underscores its commendable performance, particularly in terms of control and musicality.

## 2 RELATED WORK

### 2.1 TEXT-TO-MUSIC GENERATION

Text-based music generation using deep learning is an increasingly active research field. Various approaches have emerged. Riffusion[2] leverages stable diffusion (Rombach et al., 2022) to obtain music spectrograms paired with input texts, while Moûsai (Schneider et al., 2023) and ERNIE-Music (Zhu et al., 2023) employ diffusion models with their text-audio datasets to generate audio music. A joint embedding model, MuLan (Huang et al., 2022), links music audio and natural language descriptions, which is applied in Noise2Music (Huang et al., 2023), MusicLM (Agostinelli et al., 2023) and MeLoDy (Lam et al., 2023) for waveform generation. Also, MusicGen (Copet et al., 2023) introduces efficient token interleaving patterns to achieve audio music generation from text. Compared to audio, symbolic music, represented as musical symbols, is more manipulatable, benefiting both humans and machines in the composition process.

While there are a few studies on creating symbolic music from text, some previous attempts have limitations. For example, BUTTER (Zhang et al., 2020) focuses on generating music in ABC notations[3] from text but has limited control and language constraints. GPT-4 (OpenAI, 2023) can also generate ABC notation music but lacks complex harmony (Bubeck et al., 2023). Another study by Wu & Sun (2022) fine-tunes language models with text-music pairs but struggles to align musical attributes with text and mainly produces solo music.

In contrast, our approach, MuseCoco, offers precise control and good musical quality when generating symbolic music from text descriptions. This music format used by MuseCoco is easily editable and can be explicitly controlled using attribute values from the text, enhancing the composition process.

### 2.2 CONTROLLABLE MUSIC GENERATION

Controllable music generation means having control over specific aspects of the music being generated. Previous studies have used various models like conditional VAE (Tan & Herremans, 2020; von Rütte et al., 2023), GAN (Neves et al., 2022; Zhu et al., 2022), or diffusion models (Huang et al., 2023; Zhu et al., 2023) to achieve this. They use different conditions, such as emotions (Hung et al., 2021; Ferreira et al., 2022; Bao & Sun, 2022), styles (Mao et al., 2018; Wang et al., 2022; Choi et al., 2020), themes (Shih et al., 2022), or structures (Yu et al., 2022; Zhang et al., 2022), to control music generation. Others use music-related details like instrumentation (Ens & Pasquier, 2020; Di et al., 2021), chords (Wang et al., 2020b; Wu et al., 2022), note density (Tan & Herremans, 2020; von Rütte et al., 2023; Wu & Yang, 2021), and more to guide the music creation process.

However, prior methods had limited control over music generation, which can result in less expressive and adaptable music for users with diverse needs. These methods only controlled music based on limited attributes, limiting their ability to convey complex musical ideas. Text-based input is user-friendly and commonly used for guiding generative tasks (Rombach et al., 2022; Ramesh et al., 2022; Saharia et al., 2022; Ramesh et al., 2021; Huang et al., 2023; Agostinelli et al., 2023; Wu & Sun, 2022; Bubeck et al., 2023). Previous approaches faced difficulties in directly generating symbolic music from user-provided text descriptions because of the shortage of text-labeled data. MuseCoco offers effective solutions to address these challenges by translating text into music attributes, which are used as conditions to generate symbolic music.

---

[2]https://www.riffusion.com/about
[3]https://abcnotation.com/

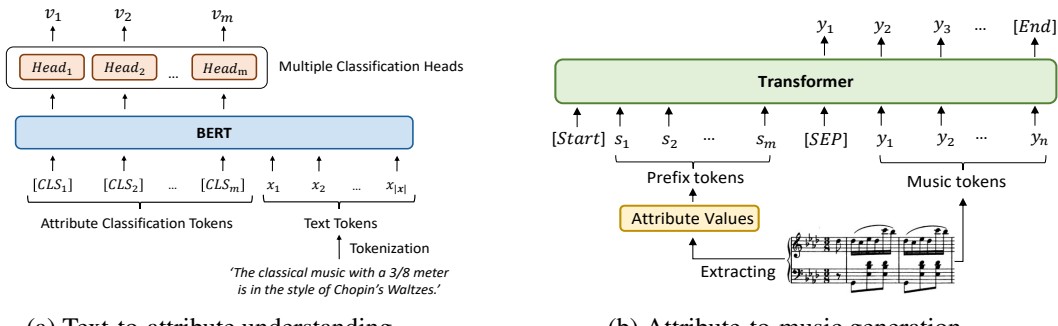

(a) Text-to-attribute understanding  (b) Attribute-to-music generation

Figure 2: The two-stage details of MuseCoco. During training, each stage is independently trained. During inference, the text-to-attribute understanding stage firstly extracts music attribute values, based on which the attribute-to-music generation stage secondly generates symbolic music.

## 3 MuseCoco

To achieve text-to-music generation, MuseCoco decomposes this task into two stages: the text-to-attribute understanding stage and the attribute-to-music generation stage. The models in these two stages are trained independently. The details of the two stages are shown in Figure 2 and we will elaborate on their technical design and model architectures in this section.

### 3.1 Text-to-attribute Understanding

Musicians usually extract or derive keywords (e.g., "low tempo", "classical style", and "piano use") from user-provided text descriptions as cues to compose. Therefore, the text-to-attribute understanding task is required to extract musical attribute values from plain text and they will be used in the later attribute-to-music generation stage to generate desired music. As shown in Figure 2(a), text-to-attribute can be denoted as $\mathcal{X} \rightarrow \mathcal{V}$, where $\mathcal{X}$ is the input text set, $\mathcal{V}$ is the value set of $m$ pre-defined musical attributes as shown in Table 1 and Appendix A. In the text-attribute dataset, each instance $\boldsymbol{x} \in \mathcal{X}$ is paired with a combination of $m$ attribute values $\boldsymbol{v} = \{v_i\}_{i=1}^{m}$, $\boldsymbol{v} \in \mathcal{V}$. Given a pre-trained language model $\mathcal{M}$, BERT$_{\text{large}}$ (Devlin et al., 2019), $\boldsymbol{x}$ is converted by the tokenizer of $\mathcal{M}$ into corresponding tokens $\{x_1, x_2, \ldots, x_{|\boldsymbol{x}|}\}$. To adapt to multiple-attribute classification, we prepend $m$ attribute classification tokens $[\text{CLS}_i]_{i=1}^{m}$ to input text tokens and each corresponding encoded $[\text{CLS}_i]$ is used to compute the probability distribution over the values of attribute $i$ with a classification head. Precisely, the input $x$ will be encoded to hidden vectors:

$$\mathbf{h}_{[\text{CLS}_1]}, \mathbf{h}_{[\text{CLS}_2]}, \ldots, \mathbf{h}_{[\text{CLS}_m]}, \mathbf{h}_{x_1}, \mathbf{h}_{x_2}, \ldots, \mathbf{h}_{x_{|\boldsymbol{x}|}}$$

The probability distribution of the value of $i$-th attribute is $p_i(v_i|\boldsymbol{x}) = \text{Softmax}(\mathbf{W_i}\mathbf{h}_{[\text{CLS}_i]} + \mathbf{b_i})$, where $\mathbf{W_i}$ and $\mathbf{b_i}$ are learnable parameters. The cross-entropy loss of $i$-th attribute is

$$\mathcal{L}_i = -\frac{1}{|\mathcal{X}|} \sum_{\boldsymbol{x} \in \mathcal{X}} \log p_i(v_i|\boldsymbol{x}) \tag{1}$$

During training, all learnable parameters are fine-tuned by minimizing the sum of attribute cross-entropy losses $\mathcal{L} = \sum_{i=1}^{m} \mathcal{L}_i$ on $\{\mathcal{X}, \mathcal{V}\}$.

### 3.2 Attribute-to-music Generation

We train the music generation model in a highly data-efficient way by leveraging large amounts of unlabeled symbolic music data since most musical attributes can be easily obtained by extracting from music sequences. Specifically, as defined in Table 1, some musical attributes are objective, about quantifiable and measurable characteristics of musical elements like tempo and meter, which can be extracted from music sequences with pre-defined rules (detailed in Section 4.1). Others are subjective, referring to musical qualities and characteristics based on personal interpretation, perception, or emotional response, like emotion and genre (Cook, 2001), which can be obtained from existing attribute-labeled datasets. Kindly be aware that the attributes mentioned herein serve as preliminary

Table 1: Musical attribute descriptions.

| Type | Attribute | Description |
|---|---|---|
| Objective | Instrument | played instruments in the music clip |
| | Pitch | the number of octaves covering all pitches in one music clip |
| | Rhythm Danceability | whether the piece sounds danceable |
| | Rhythm Intensity | the intensity of the rhythm |
| | Bar | the total number of bars in one music clip |
| | Time Signature | the time signature of the music clip |
| | Key | the tonality of the music clip |
| | Tempo | the tempo of the music clip |
| | Time | the approximate time duration of the music clip |
| Subjective | Artist | the artist (style) of the music clip |
| | Genre | the genre of the music clip |
| | Emotion | the emotion of the music clip |

examples for our initial study. We have the flexibility to expand and incorporate additional attributes to cater to a wider range of applications. Given the attributes from music sequences, we add them as prefix tokens to the music sequences. This provides explicit control over the music, making it easier to understand how the attributes influence the music. Different attribute tokens can be combined or sequenced to achieve complex musical expressions and transformations.

Precisely, attribute-to-music is denoted as $\mathcal{V} \rightarrow \mathcal{Y}$, where $\mathcal{V}$ is the set of attribute values and $\mathcal{Y}$ is the set of symbolic music. This stage transforms attribute values into prefix tokens to control music generation, as shown in Figure 2(b) since using prefix tokens to guide the generation process is an effective method for directing the output towards a particular direction (Keskar et al., 2019; Li & Liang, 2021; Liu et al., 2023; Brown et al., 2020; Wu & Yang, 2023; Dong et al., 2023; Hsiao et al., 2021). For each music sequence $\boldsymbol{y} = [y_1, y_2, ..., y_n] \in \mathcal{Y}$, and its attribute values $\boldsymbol{v} = \{v_1, v_2, ..., v_m\} \in \mathcal{V}$ in the attribute-music dataset, we consider the following distribution:

$$p(\boldsymbol{y}|\boldsymbol{v}) = \prod_{i=1}^{n} p(y_i|y_{<i}, v_1, v_2, ..., v_m). \qquad (2)$$

We encode each attribute value $v_j$ into a prefix token $s_j$. Position embeddings for such prefix tokens $\{s_j\}_{j=1}^{m}$ are not used. Since it is not usual for users to provide all attribute values in real-world scenarios, we introduce a special token, represented by $s_j^{NA}$, to exclude this attribute (i.e. $v_j$) that is not specified in inputs from influencing the music generation process. This dynamic approach allows our model to effectively adapt to a wide array of attribute distributions. Then the input sequence is encoded into:

$$s_1, s_2, s_3, ..., s_m, [\text{SEP}], y_1, y_2, ..., y_n.$$

During training, some attribute tokens (e.g., $s_2$, $s_3$) are randomly replaced with special tokens (i.e., $s_j^{NA}$) to enable adaptation to various attribute combinations. During inference, attribute values can either be sourced from extracted attributes within text inputs during the text-to-attribute understanding stage or obtained directly from pre-defined attribute values. Any attributes that are absent in the inputs are represented by special tokens, which are combined with the other prefix tokens to effectively control the music generation process as required.

### 3.3 DATA CONSTRUCTION

By using musical attributes to break down the text-to-music generation task into the text-to-attribute understanding stage and the attribute-to-music generation stage, we can leverage large amounts of symbolic music data without text descriptions. In the attribute-to-music generation stage, attributes can be extracted from music sequences with rules or obtained from attribute-labeled datasets (detailed in Section 4.1). The system only requires paired data in the text-to-attribute stage. We synthesize these text-attribute pairs in the following steps:

1. **Write templates for each attribute:** As shown in Table 2, we write several templates as a set for each attribute, where its values are represented with a placeholder. By utilizing this placeholder, we can accommodate diverse combinations of attribute values without requiring exact values.

Table 2: An example of synthesizing a text-attribute pair. We randomly choose a template for each attribute, and here are two examples for each. These templates are then improved by ChatGPT and filled in with values.

| Attribute | Value | Template |
|---|---|---|
| Key | Major | This music is composed in the [KEY] key. |
| | | This music's use of [KEY] key creates a distinct atmosphere. |
| Emotion | Happiness | The music is imbued with [EMOTION]. |
| | | The music has a [EMOTION] feeling. |
| Time Signature | 4 / 4 | The [TIME_SIGNATURE] time signature is used in the music. |
| | | The music is in [TIME_SIGNATURE]. |
| Refine via ChatGPT and fill in place-holders with values: {Key, Emotion, Time Signature} | | The music is imbued with happiness, and the major key in this music provides a powerful and memorable sound. The song progresses with 4/4 as the meter of the music. |

2. **Create attribute combinations and concatenate their templates as paired texts:** The generation process is usually controlled by multiple attributes together. Hence, constructing various different combinations of attribute values and paired text is necessary. To enrich the diversity of paired text-attribute training data and avoid the long-tailed issue in attribute distribution, we stochastically create $v$ per instance based on pre-defined musical attributes and their values on our own to ensure the number of instances including $v_i$ is balanced, i.e., each value of each attribute is sampled equally. Afterward, we create paired texts by concatenating the attribute values with randomly selected templates from sets of templates for each attribute.

3. **Refine concatenated templates via ChatGPT:** Since simply concatenated templates are less natural than real users' input, ChatGPT[4] is utilized to refine them as shown in Table 2. The provided template serves as an instructive example for reference.

4. **Fill in placeholders:** Finally, attribute values or their synonyms fill in placeholders, ensuring that the text effectively conveys the intended meaning and maintains a consistent narrative structure.

Through these steps, we can independently construct the datasets for either of the two stages without the need for paired text-music data.

## 4 EXPERIMENTS

### 4.1 EXPERIMENT SETUP

**Datasets** To train the attribute-to-music generation stage and evaluate our proposed method, we collect an assortment of MIDI datasets from online sources. Table 3 lists all of the used datasets along with their respective counts of valid MIDI files. Specifically, the MMD dataset (Zeng et al., 2021) consists of many datasets collected from the internet[5]. The EmoGen dataset (Kang et al., 2023) is generated by the emotion-controllable music generation system[5] and the others are all publicly released datasets. We did the necessary data filtering to remove duplicated and poor-quality samples, and there are 947,659 MIDI samples remaining. From each MIDI file, we randomly extracted 3 clips within 16 bars.

Table 3: Statistics of the used datasets.

| Dataset | #MIDI |
|---|---|
| MMD (Zeng et al., 2021) | 1,524,557 |
| EMOPIA (Hung et al., 2021) | 1,078 |
| MetaMidi (Ens & Pasquier, 2021) | 612,088 |
| POP909 (Wang et al., 2020a) | 909 |
| Symphony (Liu et al., 2022) | 46,360 |
| EmoGen (Kang et al., 2023) | 25,730 |
| Total (after filtering) | 947,659 |

The attributes described in Table 1 were then extracted from each clip. The objective attribute values used in the training are directly extracted from MIDI files and the subjective attribute values are obtained from some of the datasets (details in Appendix A).

---

[4]https://chat.openai.com/

[5]We obtained the dataset for this work with the help of the authors, as it was not publicly available.

**System Configuration**   In text-to-attribute understanding stage, we leverage BERT$_{\text{large}}$[6] as the backbone model and the max sequence length of it is set to 256, which covers common user input. We use 1,125 thousand samples for fine-tuning with a train/valid portion of 8:1. During training, the batch size is 64 and the learning rate is $1 \times 10^{-5}$. For the attribute-to-music generation stage, we use a REMI-like (Huang & Yang, 2020) representation method to convert MIDI into token sequences. We apply Linear Transformer (Katharopoulos et al., 2020) as the backbone model, which consists of 16 layers with causal attention and 12 attention heads. The hidden size is 1024 and FFN hidden size is 4096, yielding an approximate parameter count of 203 million. The max length of each sample is 5120, covering at most 16-bar music segments. During training, the batch size is 64. The dropout rate is set to 0.1. We use Adam optimizer (Kingma & Ba, 2014) with $\beta_1 = 0.9, \beta_2 = 0.98$ and $\epsilon = 10^{-9}$. The learning rate is $2 \times 10^{-4}$ with warm-up step 16000 and an invert-square-root decay.

**Evaluation Dataset and Metrics**   To evaluate MuseCoco, we construct a **standard test set** including 5,000 text-attribute pairs in the same way in Section 3.3. Musical attributes of each test sample originated from real music in the test set of the attribute-to-music stage, instead of creating them on our own to make sure musical rationality and all values of the attributes are covered in the test set for thorough testing. Meanwhile, in order to accord with usual user inputs, we randomly assign the *NA* value (meaning the attribute is not mentioned in the text) to some attributes per sample to synthesize text prompts with different lengths.

We calculated the text-controlled accuracy to objectively evaluate whether the attributes extracted from the generated sample align with those specified in the text descriptions. This metric calculates the proportion of correctly predicted attributes in each sample and then averages the accuracy across the entire test set. To conduct a subjective evaluation of MuseCoco's performance, we employ a user study. We invite individuals with musical backgrounds to fill out the questionnaires (details in Appendix B.1). Participants are asked to rate the following metrics on a scale of 1 (lowest) to 5 (highest): 1) **Musicality:** It assesses the degree to which the generated music exhibits qualities akin to the artistry of a human composer. 2) **Controllability:** It measures how well the samples adhere to the musical attribute values specified in the text descriptions. 3) **Overall:** It quantifies the overall quality of this generated music considering both its musicality and controllability.

## 4.2 Comparison with Baselines

**Baselines**   In this study, we compare our method to two existing works for generating symbolic music from 21 text descriptions randomly selected from the standard test set: 1) **GPT-4**: GPT-4 (OpenAI, 2023) is a large-scale language model that demonstrated its capabilities in various domains, including music. Following Bubeck et al. (2023), we instruct GPT-4 to generate ABC notation music with the task-specific prompts (in Appendix B.5). 2) **BART-base**: Wu & Sun (2022) release a language-music BART-base[7], which shows a solid performance. Text descriptions are fed into this model and guide it to generate ABC notation music for comparison.

These two end-to-end works are representative of generating symbolic music from text. Comparing MuseCoco's performance against these leading models provides a strong and convincing showcase of its effectiveness. As for the subjective evaluation, well-designed questionnaires including generated music from baselines and our method are distributed to individuals, who are all in music backgrounds and required to score the subjective metrics described in Section 4.1 (details in Appendix B.1). Meanwhile, to objectively compare the model ability, we calculate the average sample-wise accuracy of generated music for both baselines and our method (details in Appendix B.2).

**Main Results**   Table 4 reports the main results of the comparison. MuseCoco excels in musicality with a mean score of 4.06 out of 5, closely resembling human compositions and real-world music. It outperforms baselines in conditional generation, scoring 35.2% in controllability and 19.95% in sample-wise accuracy, showcasing effective control through its two-stage framework. MuseCoco maintains the best overall score at 4.13 out of 5 in auditory tests, indicating preferred and high-quality music generation. End-to-end models, while theoretically information-retentive, lag behind MuseCoco in control, musicality, output, and text-controlled accuracy, revealing susceptibility to information loss. MuseCoco stands out across all metrics, overcoming end-to-end model limitations,

---

[6]https://huggingface.co/bert-large-uncased
[7]https://huggingface.co/sander-wood/text-to-music

leveraging abundant training data without the need for paired text-symbolic music data, offering clear attribute definitions, and prioritizing interpretability for user-driven music generation.

Table 4: Comparison between MuseCoco, GPT-4, and BART-base. Acc. (text) stands for the text-controlled accuracy.

|  | Musicality ↑ | Controllability ↑ | Overall ↑ | Acc. (text) (%) ↑ |
|---|---|---|---|---|
| MuseCoco | **4.06** ± 0.82 | **4.15** ± 0.78 | **4.13** ± 0.75 | **77.59** |
| GPT-4 (OpenAI, 2023) | 2.79 ± 0.97 | 3.07 ± 1.05 | 2.81 ± 0.97 | 57.64 |
| BART-base (Wu & Sun, 2022) | 2.19 ± 1.14 | 2.02 ± 1.09 | 2.17 ± 1.03 | 32.47 |

### 4.3 METHOD ANALYSIS

In this section, we conduct analysis experiments on the two stages respectively.

#### 4.3.1 ANALYSIS ON TEXT-TO-ATTRIBUTE UNDERSTANDING

**Attribute Comprehension**    To evaluate the ability to extract each attribute from text, we test the text-to-attribute model on the standard test set and show the classification accuracy of each attribute in Appendix B.3. Each accuracy consistently surpasses 99%, indicating that the model has exceptional performance on all the attributes and is reliable on text understanding.

**Different Classification Heads**    We explore the effectiveness of using multiple classification heads and report the text-controlled accuracy. The model with multiple classifications shows 99.96% text-controlled accuracy, while the one-head BERT reports 60.09% text-controlled accuracy. The 39.87% difference illustrates that each head can learn their corresponding attribute knowledge, and using multiple heads can improve the overall performance.

**Generalization**    To illustrate our system's ability to accommodate diverse language patterns employed by different users, we extended invitations to four musicians and they crafted 17 text descriptions, allowing us to conduct controll accuracy evaluations. Our system achieves an impressive 94.96% text-controlled accuracy score, which is highly promising. Furthermore, it's worth noting that while the text descriptions were synthesized for training purposes, our system has demonstrated its capacity to adapt to real-world language usage scenarios with agility and proficiency.

#### 4.3.2 ANALYSIS ON ATTRIBUTE-TO-MUSIC GENERATION

**Attribute Control Performance**    To evaluate the controllability of the attribute-to-music generation model, we report the control accuracy results for each attribute in Appendix B.4. The average attribute-controlled accuracy is 80.42%, demonstrating a strong capability of the model to effectively respond to the specified attribute values during the music generation process.

**Study on Control Methods**    We compare *Prefix Control*, which is the default method of our model that uses prefix tokens to control music generation, with two other methods: 1) *Embedding*: Add attribute input as embedding to token embedding (Wu & Yang, 2021); 2) *Conditional LayerNorm*: Add attribute input as a condition to the layer norm layer (Perez et al., 2018; Chen et al., 2021). We utilize Musicality and average attribute control accuracy as evaluation metrics. For more details on this experiment, please refer to the description in Appendix B.4. We report evaluation results in Table 5. It is shown that *Prefix Control* outperforms other methods in terms of musicality and average attribute control accuracy, with a minimum improvement of 1.29% and 19.94% respectively, highlighting its superior capability to capture the relationship between attributes and music.

**Study on Model Size**    We conduct a comparative analysis between two different model sizes to determine whether increasing the model size would result in improved generated results. The parameter configurations for these model sizes are presented in Table 6. The model, referred to as

Table 5: Comparison of different control methods. Musicality reflects the quality of the generated music. Acc.(attribute) stands for average attribute-controlled accuracy, which represents the control accuracy over all attributes to reflect controllability.

| Method | Musicality ↑ | Acc.(attribute) (%) ↑ |
|---|---|---|
| Embedding | $2.97 \pm 0.91$ | 36.94 |
| Conditional LayerNorm | $3.11 \pm 1.02$ | 47.46 |
| Prefix Control | $\mathbf{3.15} \pm 1.02$ | **67.40** |

Table 6: Comparison of different model sizes in the attribute-to-music generation stage.

| Model Size | Layers | $d_{model}$ | Parameters | Musicality ↑ | Acc.(attribute) (%) ↑ |
|---|---|---|---|---|---|
| large | 16 | 1024 | 203M | 2.80 | 70.69 |
| xlarge | 24 | 2048 | 1.2B | **3.22** | **87.23** |

*large*, is the default model for the attribute-to-music generation stage. Additionally, we utilize *xlarge* model for comparison, which consists of approximately 1.2 billion parameters. The training of *xlarge* model follows the same settings outlined in Section 4.1. The evaluation results are displayed in Table 6, which indicates that increasing the model size enhances controllability and musicality.

### 4.3.3 COMMENTS FROM MUSICIANS

We invite professional musicians to give their comments on generated music samples from given texts based on our system. The feedback from musicians has demonstrated our ability to enhance their workflow efficiency by reducing redundant tasks and providing creative inspiration as the following:

From Musician A: *The generated music is remarkably similar to human compositions, showing impressive accuracy and creativity. It inspires creative work with intriguing motifs and skillful organization of musical elements. This significantly speeds up composition for about one day of time.*

From Musician B: *The generated music offers arrangement inspiration, exemplified by the idea of combining right-hand arpeggios and left-hand bass melody to facilitate creative expansion in composition. It sparks the concept of blending classical and popular music genres described in texts.*

From Musician C: *The generated music has substantially sped up my composition process, saving me 2 days to 2 weeks, especially for unfamiliar instrumental arrangements. The composition includes inspiring sections, like the journey through conflicts and resolutions. In some areas towards the end it felt like I was embarking on an adventure up a mountain and through grassy fields, very interesting.*

## 5 DISCUSSION

**Conclusion** This paper presents MuseCoco, a system that transforms text descriptions into coherent symbolic compositions, benefiting musicians and users of varied backgrounds. Our two-stage design simplifies learning, reducing reliance on extensive text-music data and offering precise control through attributes. By leveraging a substantial volume of training data (approximately one million instances), we ensure comprehensive coverage of diverse attribute combinations in textual inputs, enhance musicality and text-music coherence. This research demonstrates AI's potential in aiding music creation, offering MuseCoco as a versatile tool to inspire artists and streamline composition.

**Future Work** This study is an early effort to demonstrate creating symbolic music from text descriptions without needing labeled data pairs, potentially simplifying music composition. However, we've only explored a portion of musical attributes. In the future, we aim to explore new attributes like temporal information and semantic control for more creative control and diversity in music generation. While symbolic notation is editable, it lacks timbral expressiveness. To improve this, we can add a timbre synthesis module, empowering musicians to make better decisions during composition.

ETHICS STATEMENT

When involving human participants in our research, we ensure that they provide informed consent. This includes explaining the purpose of the study, how their data will be used, and any potential risks involved. We are mindful of copyright and intellectual property rights in our work. We strive to create original compositions and respect the rights of copyright holders. Any use of copyrighted material is done in accordance with applicable laws and regulations. However, the use of generative AI in creative contexts raises copyright and ownership concerns that warrant careful consideration. We view our work in automatic music generation as a means of enhancing human creativity and collaboration with AI, rather than a replacement for human musicians and composers.

REPRODUCIBILITY

We have made our source code available in Supplementary Materials for the purpose of replicating the data processing, model training, and inference processes. However, due to copyright constraints, we are unable to release all the datasets used. Nevertheless, we will provide checkpoints that can be used to reproduce the results. Sections Section 3.1, Section 3.2, and Section 3.3 offer an overview of the system's design principles, while Section Section 4.1 provides specific details about system configurations. For more comprehensive information on attributes, the utilization of GPT models, and data processing techniques, please refer to the Appendix.

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

## A  ATTRIBUTE INFORMATION

Table 7 shows the detailed pre-defined musical attribute values. The value *NA* of each attribute refers to that this attribute is not mentioned in the text. Objective attributes can be extracted from MIDI files with heuristic algorithms and subjective attributes are collected from existing datasets, as shown in Table 8.

Table 7: Detailed attribute values.

| Attributes | Values |
|---|---|
| Instrument | 28 instruments: piano, keyboard, percussion, organ, guitar, bass, violin, viola, cello, harp, strings, voice, trumpet, trombone, tuba, horn, brass, sax, oboe, bassoon, clarinet, piccolo, flute, pipe, synthesizer, ethnic instrument, sound effect, drum. Each instrument: 0: played, 1: not played, 2: NA |
| Pitch Range | 0-11: octaves,12: NA |
| Rhythm Danceability | 0: danceable, 1: not danceable, 2: NA |
| Rhythm Intensity | 0: serene, 1: moderate, 2: intense, 3: NA |
| Bar | 0: 1-4 bars, 1: 5-8 bars, 2: 9-12 bars, 3: 13-16 bars, 4: NA |
| Time Signature | 0: 4/4, 1: 2/4, 2: 3/4, 3: 1/4, 4: 6/8, 5: 3/8, 6: other tempos, 7: NA |
| Key | 0: major, 1: minor, 2: NA |
| Tempo | 0: slow (<=76 BPM), 1: moderato (76-120 BPM), 2: fast (>=120 BPM), 3: NA |
| Time | 0: 0-15s, 1: 15-30s, 2: 30-45s, 3: 45-60s, 4: >60s, 5: NA |
| Artist | 0-16 artists: Beethoven, Mozart, Chopin, Schubert, Schumann,J.S.Bach, Haydn, Brahms, Handel, Tchaikovsky, Mendelssohn,Dvorak, Liszt, Stravinsky, Mahler,Prokofiev, Shostakovich, 17: NA |
| Genre | 21 genres: new age, electronic, rap, religious,international,easy listening, avant garde, RNB, latin, children, jazz, classical, comedy, pop, reggae, stage, folk, blues, vocal, holiday, country, symphony Each genre: 0: with, 1: without, 2: NA |
| Emotion | 0-3: the 1-4 quadrant in Russell's valence-arousal emotion space 4: NA |

Table 8: Extraction methods and sources of each attributes.

| Type | Attribute | Extraction Method |
|---|---|---|
| Objective | Instrument | directly extracted from MIDI |
| | Pitch range | calculated based on the pitch range |
| | Rhythm danceability | judged with the ratio of downbeat |
| | Rhythm intensity | judged with the average note density |
| | Bar | directly extracted from MIDI |
| | Time signature | directly extracted from MIDI |
| | Key | judged with the note pitches based on musical rules |
| | Tempo | directly extracted from MIDI |
| | Time | derived from the time signature and the number of bars |
| Subjective | Artist | provided by a classical music dataset in MMD (Zeng et al., 2021) |
| | Genre | provided by MAGD[8], a classical music dataset in MMD (Zeng et al., 2021) and Symphony (Liu et al., 2022) |
| | Emotion | provided by EMOPIA (Hung et al., 2021) and the emotion-gen dataset |

## B  EXPERIMENTS

### B.1  USER STUDY WITH BASELINES

In the user study, participants were provided with generated music samples along with their corresponding textual prompts. For each text description, each model (i.e., BART-base, GPT-4, MuseCoco) generated three different music clips. In each questionnaire, three samples generated with the same

text conditions were randomly picked from samples generated by BART-base, GPT-4, and MuseCoco respectively as a group. Each participant was asked to evaluate 7 groups for comparison. Three subjective metrics, musicality, controllability, and an overall score, are rated on a scale of 1 (lowest) to 5 (highest). The participants were first requested to evaluate their music profession level, as depicted in Table 9. To ensure the reliability of the assessment, only individuals with at least music profession level 3 were selected, resulting in a total of 19 participants. Secondly, they were instructed to independently evaluate two separate metrics: musicality and controllability, ensuring that scoring for one metric did not influence the other. They are also asked to give an overall score to evaluate the generated music comprehensively. We asked the following questions according to each metric in the questionnaire: 1) To evaluate musicality, we asked them to evaluate whether the music is pleasant, melodious, smooth, interesting, and approaches human-created music. 2) To evaluate controllability, we asked them to assess whether the music meets the requirements described in the text descriptions. 3) To evaluate the overall quality, we asked them to give their overall score interms of both musicality and controllability. The details of each level we defined for each metric is illustrated in Table 10 for musicality, Table 11 for controllability, and Table 12 for overall quality. For the collected results, we computed the mean and variance for each metric. The results can be found in Table 4.

Table 9: Music Profession Level

| Level | Description |
|---|---|
| 1 | I rarely listen to music. |
| 2 | I haven't received formal training in playing or music theory, but I often listen to music and have my preferred styles, musicians, and genres. |
| 3 | I have some basic knowledge of playing an instrument or music theory, but I haven't received formal training. |
| 4 | I haven't received formal training, but I have self-taught myself some aspects such as music theory or playing an instrument. I am at an amateur level (e.g., CCOM piano level 6 or above). |
| 5 | I have received professional training in a systematic manner. |

Table 10: Musicality Level

| Level | Description |
|---|---|
| 1 | Not even close. |
| 2 | Slightly approaching. |
| 3 | Moderately approaching. |
| 4 | Relatively close. |
| 5 | Extremely approaching, or it is human-composed music. |

Table 11: Controllability Level

| Level | Description |
|---|---|
| 1 | Not in the slightest accordance. |
| 2 | Somewhat in accordance. |
| 3 | Generally meets. |
| 4 | Relatively in accordance. |
| 5 | Completely in accordance. |

## B.2 OBJECTIVE COMPARISON WITH BASELINES

In this section, we introduce how to calculate the objective metric, the text-controlled accuracy, in Table 4. As for MuseCoco, ten music clips are generated per prompt and we report text-controlled

Table 12: Overall Quality Level

| Level | Description |
|-------|-------------|
| 1 | Very poor. |
| 2 | Relatively poor. |
| 3 | Average proficiency. |
| 4 | Quite good. |
| 5 | Excellent. |

accuracy of them among the overall standard test set. Since it is labor-intensive to leverage GPT-4 with the official web page, we only guide GPT-4 to produce five music clips per prompt and calculate the text-controlled accuracy of 21 prompts randomly sampled from the standard test set. Besides, we utilize the released text-tune BART-base checkpoint[9] to generate five music clips per prompt and report the text-controlled accuracy of 44 prompts randomly chosen from the standard test set.

### B.3 TEXT-TO-ATTRIBUTE UNDERSTANDING

As shown in Table 13, all attribute control accuracy is close or equal to 100%, which indicates our model with multiple classification heads in the text-to-attribute understanding stage performs quite well.

Table 13: Attribute control accuracy (%) of the text-to-attribute understanding model. I: Instrument.

| Attribute | Accuracy(%) | Attribute | Accuracy(%) | Attribute | Accuracy(%) |
|-----------|-------------|-----------|-------------|-----------|-------------|
| I_piano | 100.00 | I_clarinet | 99.92 | Genre_comedy_spoken | 100.00 |
| I_keyboard | 99.92 | I_piccolo | 99.94 | Genre_pop_rock | 100.00 |
| I_percussion | 100.00 | I_flute | 99.62 | Genre_reggae | 100.00 |
| I_organ | 100.00 | I_pipe | 100.00 | Genre_stage | 100.00 |
| I_guitar | 99.92 | I_synthesizer | 100.00 | Genre_folk | 100.00 |
| I_bass | 99.84 | I_ethnic_instruments | 99.98 | Genre_blues | 100.00 |
| I_violin | 99.92 | I_sound_effects | 99.98 | Genre_vocal | 100.00 |
| I_viola | 99.96 | I_drum | 100.00 | Genre_holiday | 100.00 |
| I_cello | 99.92 | Genre_new_age | 99.98 | Genre_country | 100.00 |
| I_harp | 100.00 | Genre_electronic | 100.00 | Genre_symphony | 100.00 |
| I_strings | 99.96 | Genre_rap | 100.00 | Bar | 100.00 |
| I_voice | 99.70 | Genre_religious | 100.00 | Time Signature | 100.00 |
| I_trumpet | 99.96 | Genre_international | 100.00 | Key | 100.00 |
| I_trombone | 99.94 | Genre_easy_listening | 100.00 | Tempo | 99.84 |
| I_tuba | 100.00 | Genre_avant_garde | 100.00 | Octave | 100.00 |
| I_horn | 99.94 | Genre_rnb | 100.00 | Emotion | 99.80 |
| I_brass | 100.00 | Genre_latin | 100.00 | Time | 100.00 |
| I_sax | 99.84 | Genre_children | 100.00 | Rhythm Danceability | 100.00 |
| I_oboe | 99.94 | Genre_jazz | 100.00 | Rhythm Intensity | 99.88 |
| I_bassoon | 99.96 | Genre_classical | 100.00 | Artist | 100.00 |

### B.4 DETAILS OF ANALYSIS ON ATTRIBUTE-TO-MUSIC GENERATION

**Attribute Control Accuracy** We report the control accuracy for each attribute on the test dataset, as shown in Table 14. The average attribute control accuracy of 80.42%, which provides substantial evidence for the model's proficiency in effectively controlling music generation using music attributes.

**Study on Control Methods** To verify the effectiveness of the control method in the attribute-to-music generations stage, we compare *Prefix Control* with two methods: *Embedding* and *Conditional LayerNorm*. For efficiency, we conducted this study on reduced-size models as follows: The backbone

---
[9] https://huggingface.co/sander-wood/text-to-music

Table 14: Accuracy (%) of each attribute for attribute-to-music generation. I: Instrument.

| Attribute | Accuracy(%) | Attribute | Accuracy(%) |
|---|---|---|---|
| I_piano | 96.20 | I_clarinet | 90.63 |
| I_keyboard | 79.55 | I_piccolo | 86.67 |
| I_percussion | 65.19 | I_flute | 86.73 |
| I_organ | 80.55 | I_pipe | 70.73 |
| I_guitar | 91.81 | I_synthesizer | 78.28 |
| I_bass | 93.11 | I_ethnic_instruments | 77.69 |
| I_violin | 87.88 | I_sound_effects | 51.74 |
| I_viola | 92.03 | I_drum | 95.96 |
| I_cello | 86.50 | Bar | 71.80 |
| I_harp | 74.87 | Time Signature | 99.14 |
| I_strings | 86.08 | Key | 57.42 |
| I_voice | 75.82 | Tempo | 92.71 |
| I_trumpet | 84.86 | Octave | 61.56 |
| I_trombone | 84.64 | Time | 65.82 |
| I_tuba | 93.08 | Rhythm Danceability | 88.04 |
| I_horn | 80.13 | Rhythm Intensity | 80.47 |
| I_brass | 77.27 | Genre | 73.08 |
| I_sax | 81.74 | Emotion | 69.45 |
| I_oboe | 85.23 | Artist | 50.03 |
| I_bassoon | 90.72 | | |

model of this experiment is a 6-layer Linear Transformer with causal attention. The hidden size is 512 and the FFN hidden size is 2048. The other experiment configuration is the same as Section 4.1. Since the control accuracy of objective attributes can be easily calculated, we only need to measure the controllability of each subjective attribute in listening tests. The control accuracy of each attribute is shown in Table 14. Finally, the average attribute control accuracy can be calculated based on the accuracy results from both types of attributes. To measure the controllability of subjective attributes (such as emotion and genre), we invite 12 participants to conduct a listening test. Each participant was provided with 18 music pieces (6 pieces per control method) with corresponding subjective attributes. We asked each participant to answer: 1) Musicality(five-point scale): How similar it sounds to the music composed by a human. 2) Controllability: Does it align with the given attributes? Then we report the musicality and average attribute accuracy score in Table Table 5. The experimental results clearly demonstrate that *Prefix Control* outperforms the other two methods in terms of musicality and controllability.

### B.5 USAGE OF GPT MODELS

**Refine texts with ChatGPT** As shown in Figure 3, in order to make text descriptions more coherent and fluent, we feed concatenated templates into ChatGPT with a prompt *Please combine the following sentences to one paragraph* and then ChatGPT will give a response containing all templates within a compact paragraph.

**Generate ABC notation music with GPT-4** To use GPT-4 as the baseline method for comparison, we design the instruction to guide GPT-4 as shown in Figure 4. GPT-4 can only generate symbolic music in ABC notation, so we need to explicitly point out the format. Besides, since GPT-4 can generate various ABC notation formats, some of which cannot be processed by music21, we provide an ABC notation example, teaching GPT-4 to follow its format. Meanwhile, we use the prompt, *Compose ABC notation music as the format above according to the music description as follows: [text descriptions]* to let GPT-4 generate music according to the text description. And we finally convert the ABC notations into MIDI for a fair comparison.

**Prompt**

Please combine the following sentences to one paragraph:
The music is imbued with [EMOTION]. The [KEY] key in this music provides a powerful and memorable sound. The song progresses through [NUM_BARS] bars. [TIME_SIGNATURE] is the meter of the music.

**ChatGPT Response**

The music is imbued with [EMOTION], creating a captivating and emotionally rich experience. It is composed in the [KEY] key, which adds a powerful and memorable sound to the composition. The song unfolds through [NUM_BARS] bars, with [TIME_SIGNATURE] serving as the meter, further enhancing its rhythmic and melodic structure.

Figure 3: ChatGPT refines concatenated templates in the prompt.

**Prompt**

I want a music in the ABC notation format and assign proper MIDI instrument IDs as the following:

X:1
T:Forest Stroll
M:3/4
L:1/4
Q:1/4=80
K:C
V:1
%%MIDI program 90 % Synthesizer
E/2F/2|G2A|G2F|E2D|C2E|F2G|A3-|A2E/2F/2|
G2A|G2F|E2D|C2E|F2G|A2F|E2D|C3||
V:2
%%MIDI program 68 % Oboe
G/2A/2|B2C|B2A|G2F|E2G|A2B|c3-|c2G/2A/2|
B2C|B2A|G2F|E2G|A2B|c2A|G2F|E3||
V:3
%%MIDI program 73 % Flute
z/2A/2|B2C|B2A|G2F|E2G|A2B|c3-|c2z/2A/2|
B2C|B2A|G2F|E2G|A2B|c2A|G2F|E3||
V:4
%%MIDI program 42 % Cello
C/2D/2|E2F|E2D|C2B,|A,2C|D2E|F3-|F2C/2D/2|
E2F|E2D|C2B,|A,2C|D2E|F2D|C2B,|A,3||

Compose ABC notation music as the format above according to the music description as follow: The music should feature the cello and flute. The music is in 4/4 meter. The song is 31 ~ 45 seconds in length.

**GPT-4 Response**

X:1
T:Whispering Winds
M:4/4
L:1/4
Q:1/4=80
K:C
V:1
%%MIDI program 73 % Flute
A/2B/2|c2d|e2c|A2G|F2A|B2c|d3-|d2A/2B/2|
c2d|e2c|A2G|F2A|B2c|d2A|G2F|A3||
V:2
%%MIDI program 42 % Cello
C/2E/2|F2G|A2F|C2B,|A,2C|E2F|G3-|G2C/2E/2|
F2G|A2F|C2B,|A,2C|E2F|G2C|B,2A,|C3||

**Whispering Winds**

Figure 4: GPT-4 generates ABC notation tunes based on the prompt.

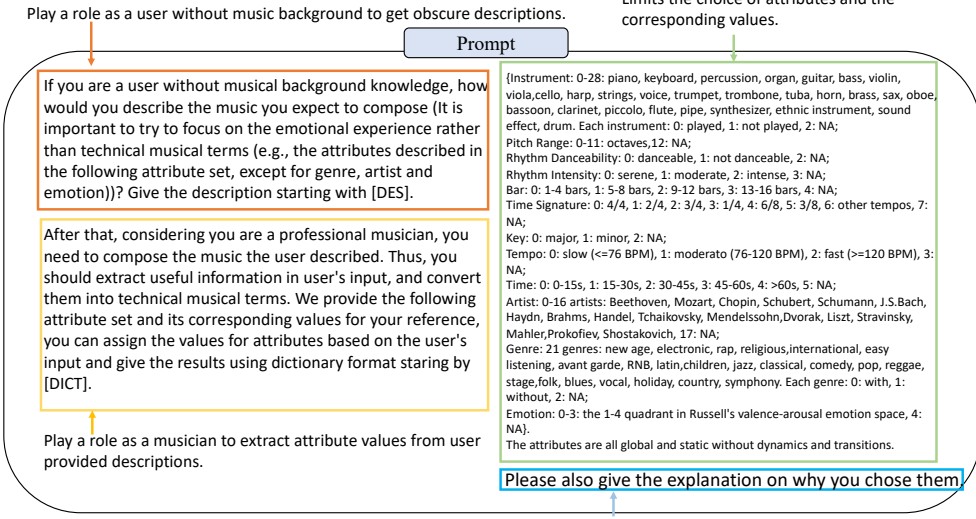

Play a role as a user without music background to get obscure descriptions.

Limits the choice of attributes and the corresponding values.

**Prompt**

If you are a user without musical background knowledge, how would you describe the music you expect to compose (It is important to try to focus on the emotional experience rather than technical musical terms (e.g., the attributes described in the following attribute set, except for genre, artist and emotion))? Give the description starting with [DES].

After that, considering you are a professional musician, you need to compose the music the user described. Thus, you should extract useful information in user's input, and convert them into technical musical terms. We provide the following attribute set and its corresponding values for your reference, you can assign the values for attributes based on the user's input and give the results using dictionary format staring by [DICT].

Play a role as a musician to extract attribute values from user provided descriptions.

{Instrument: 0-28: piano, keyboard, percussion, organ, guitar, bass, violin, viola, cello, harp, strings, voice, trumpet, trombone, tuba, horn, brass, sax, oboe, bassoon, clarinet, piccolo, flute, pipe, synthesizer, ethnic instrument, sound effect, drum. Each instrument: 0: played, 1: not played, 2: NA;
Pitch Range: 0-11: octaves,12: NA;
Rhythm Danceability: 0: danceable, 1: not danceable, 2: NA;
Rhythm Intensity: 0: serene, 1: moderate, 2: intense, 3: NA;
Bar: 0: 1-4 bars, 1: 5-8 bars, 2: 9-12 bars, 3: 13-16 bars, 4: NA;
Time Signature: 0: 4/4, 1: 2/4, 2: 3/4, 3: 1/4, 4: 6/8, 5: 3/8, 6: other tempos, 7: NA;
Key: 0: major, 1: minor, 2: NA;
Tempo: 0: slow (<=76 BPM), 1: moderato (76-120 BPM), 2: fast (>=120 BPM), 3: NA;
Time: 0: 0-15s, 1: 15-30s, 2: 30-45s, 3: 45-60s, 4: >60s, 5: NA;
Artist: 0-16 artists: Beethoven, Mozart, Chopin, Schubert, Schumann, J.S.Bach, Haydn, Brahms, Handel, Tchaikovsky, Mendelssohn, Dvorak, Liszt, Stravinsky, Mahler, Prokofiev, Shostakovich, 17: NA;
Genre: 21 genres: new age, electronic, rap, religious, international, easy listening, avant garde, RNB, latin, children, jazz, classical, comedy, pop, reggae, stage, folk, blues, vocal, holiday, country, symphony. Each genre: 0: with, 1: without, 2: NA;
Emotion: 0-3: the 1-4 quadrant in Russell's valence-arousal emotion space, 4: NA}.
The attributes are all global and static without dynamics and transitions.

Please also give the explanation on why you chose them.

Explanations are required to ensure the stability.

Figure 5: Prompt for ChatGPT to generate obscure descriptions.

**Input text descriptions**

[DES] I'd like music that feels like a calm, peaceful morning. Something that could be played in the background as I'm sipping my coffee and reading a book. I imagine it would be very serene and soothing, not too fast or upbeat. I don't want it to be danceable, but rather something I can just relax to. I think a piano would be nice, maybe with some soft strings or a flute in the background.

**Output attributes and values**

[DICT] {'Instrument: 0-28': {'piano': 0, 'strings': 0, 'flute': 0, 'others': 2}, 'Pitch Range': 5, 'Rhythm Danceability': 1, 'Rhythm Intensity': 0, 'Bar': 2, 'Time Signature': 0, 'Key': 0, 'Tempo': 0, 'Time': 4, 'Artist': 17, 'Genre': {'new age': 0, 'others': 2}, 'Emotion': 0}

**Explanation**

EXPLANATION: The user described wanting calm and peaceful music, which aligns with the 'serene' option for Rhythm Intensity and the 'slow' option for Tempo. They mentioned not wanting it to be danceable, so Rhythm Danceability was set to 'not danceable'. They also mentioned piano, strings, and flute, so those were set to 'played', while all other instruments were set to 'NA'. Because they didn't specify a particular artist, genre, or key, those were set to 'NA', 'new age' (as it often has calm and peaceful music), and 'major' (which generally sounds happy and serene). The time was set to '>60s' because they mentioned wanting to listen while reading a book, which typically takes more than a minute. And finally, because the music is described as serene and soothing, the Emotion was set to the first quadrant in Russell's valence-arousal emotion space, which represents high valence, low arousal emotions such as serene, content, and satisfied.

Figure 6: Response from ChatGPT to generate obscure descriptions and selected attribute values.

[DES] I want the music to feel like a peaceful morning, like when you're sitting by a window and watching the sunrise. It should evoke a sense of calm and tranquility, like a soft whisper of the wind or the gentle rustling of the leaves. I don't want anything too loud or intense, just something light and soothing that can make me feel relaxed and at peace.

[DICT] {'Instrument': {0: 1, 1: 2, 2: 1, 3: 1, 4: 0, 5: 2, 6: 1, 7: 1, 8: 2, 9: 0, 10: 1, 11: 2, 12: 2, 13: 2, 14: 2, 15: 2, 16: 2, 17: 2, 18: 2, 19: 2, 20: 2, 21: 2, 22: 2, 23: 0, 24: 2, 25: 1, 26: 2, 27: 2, 28: 2}, 'Pitch Range': 4, 'Rhythm Danceability': 1, 'Rhythm Intensity': 0, 'Bar': 0, 'Time Signature': 0, 'Key': 0, 'Tempo': 0, 'Time': 4, 'Artist': 17, 'Genre': {0: 2, 1: 2, 2: 2, 3: 2, 4: 2, 5: 0, 6: 2, 7: 2, 8: 2, 9: 2, 10: 2, 11: 2, 12: 2, 13: 2, 14: 2, 15: 2, 16: 2, 17: 2, 18: 2, 19: 2, 20: 2, 21: 2}, 'Emotion': 0}

Explanation: Based on the user's description, the music should be calm and peaceful. Hence, the instruments used are guitar and harp, which are known for producing soothing sounds. The pitch range is set to 4, indicating a moderate pitch that is neither too high nor too low. The rhythm is not danceable and serene in intensity, reflecting tranquility. The time signature is 4/4, a common time for calm music. The key is major, which often sounds happy and relaxed. The tempo is slow, providing a relaxed pace. The music lasts for more than 60 seconds to provide ample time for the listener to relax and enjoy. The genre is set to easy listening, which is designed to be soothing and relaxing. Finally, the emotion quadrant in Russell's valence-arousal emotion space is set to 0, corresponding to a calm and relaxed state.

(a) Example 1.

[DES] I imagine a piece of music that makes me feel enthusiastic and brings a feeling of adventure and excitement. It should have a fast pace like a foot-race, a high energy level that makes you want to move and dance. I would like to hear some strong beats like drum and bass to emphasise the rhythm. Also, I prefer the use of electronic instruments, something like a synthesizer would be great.

[DICT] {'Instrument': {0: 2, 1: 2, 2: 2, 3: 2, 4: 2, 5: 2, 6: 2, 7: 2, 8: 2, 9: 2, 10: 2, 11: 2, 12: 2, 13: 2, 14: 2, 15: 2, 16: 2, 17: 2, 18: 2, 19: 2, 20: 2, 21: 2, 22: 2, 23: 2, 24: 2, 25: 0, 26: 2, 27: 0, 28: 0}, 'Pitch Range': 12, 'Rhythm Danceability': 0, 'Rhythm Intensity': 2, 'Bar': 2, 'Time Signature': 0, 'Key': 0, 'Tempo': 2, 'Time': 4, 'Artist': 17, 'Genre': {0: 1, 1: 0, 2: 1, 3: 1, 4: 1, 5: 1, 6: 1, 7: 1, 8: 1, 9: 1, 10: 1, 11: 1, 12: 1, 13: 1, 14: 1, 15: 1, 16: 1, 17: 1, 18: 1, 19: 1, 20: 1, 21: 1}, 'Emotion': 2}

Explanation: The user mentioned that they want to feel enthusiastic and excited which leads to the emotion quadrant 2 in Russell's valence-arousal emotion space. They mentioned a fast pace, which corresponds to a fast tempo (>=120 BPM). They also mentioned strong beats and high energy level, which implies a danceable rhythm and intense rhythm intensity. The usage of electronic instruments such as synthesizer and drum indicates genre as electronic. The mention of adventure suggests a major key, as it often conveys happy and bright emotions. The user did not specify any artist, so the artist is NA. The user did not specify the pitch range, bar, or time, so these attributes are NA.

(b) Example 2.

Figure 7: More examples of response to obscure descriptions.

## C   CONVERTING ABC NOTATION TO MIDI

We transform the ABC notation music generated by the baselines into MIDI music using the music21 library [10]. After getting the midi files, the same procedure for the extraction of attributes follows the rules outlined in Table 8. Elements such as instrument, bar, time signature, and tempo are directly derived from MIDI sequences. The pitch range is computed by subtracting the minimum pitch from the maximum pitch within a music sequence. Rhythm danceability is determined by calculating the ratio of downbeats, while rhythm density is measured as the average note density. Key estimation is based on note distribution using musical rules. Additionally, time is derived from the time signature and the total number of bars.

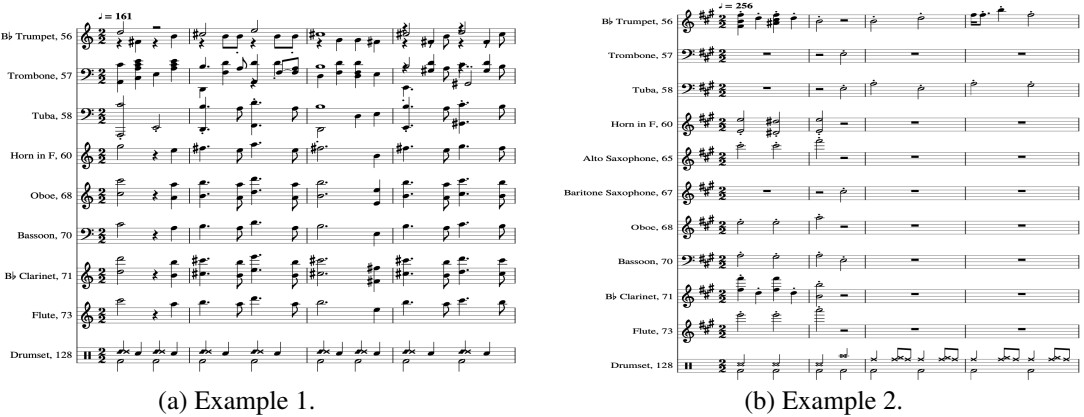

(a) Example 1.                                        (b) Example 2.

Figure 8: More example lead sheets of generated music.

## D   EXTENSION TO OBSCURE DESCRIPTIONS

In the process of selecting attributes as conditions, we carefully considered the inclusion of explicit and obscure descriptions to cater to diverse user preferences. For professional musicians, efficiency is enhanced by providing technical music terms, enabling the model to generate music explicitly aligned with their descriptions. Conversely, users without a music background express their ideas using more obscure language to convey emotions or preferred styles. Consequently, we utilize objective attributes to streamline the music creation process for musicians and introduce subjective attributes such as emotion, artistic style, and genres to facilitate expressive music generation for users of varying backgrounds and preferences.

Furthermore, our framework is designed to be expansible, allowing for the inclusion of more intriguing and diverse descriptions. We endeavored to construct an additional dataset by leveraging ChatGPT to fine-tune the text-to-attribute understanding stage. In this phase, we instructed ChatGPT to simulate the role of a musician, randomly selecting attributes and their corresponding values (which can also be user-provided). Subsequently, we tasked ChatGPT, assuming the role of a user without a music background, to employ these chosen values in generating descriptions enriched with more abstract language. To ensure logical alignment, we let it provide an explanation of how the description corresponds to the selected values. The details of the prompts is shown in Figure 5 and an example of response from ChatGPT is shown in Figure 6. More example responses are shown in Figure 7.

In our efforts to extend our framework to encompass a broader range of text descriptions, we discovered the significance of prompt engineering in constructing paired text-to-attribute data. Experimenting with more meticulously designed prompts is imperative. This can indicate potential future directions to advance the development of this field.

---

[10]http://web.mit.edu/music21/

## E  EXAMPLE LEAD SHEETS OF GENERATED MUSIC

In this section, we give some examples generated by MuseCoco in Figure 8 with the following text description:

Music is representative of the typical pop sound and spans 13 to 16 bars, this is a song that has a bright feeling from the beginning to the end. This song has a very fast and lively rhythm. The use of a specific pitch range of 3 octaves creates a cohesive and unified sound throughout the musical piece.

Due to the page limit, we only show first previous bars, please refer to the demo page for a complete version.

## F  DISCUSSION ON ATTRIBUTES SELECTION

The chosen attributes are grounded in musical theory and draw inspiration from prior research. Leveraging JSymbolic [11], a widely-used tool for extracting statistical information from music data in the field of Music Information Retrieval (MIR), we ensure that each category we utilize features at least one attribute. The representation follows the format "attribute-category" to signify the coverage within each category. Examples include pitch range, key-pitch statistics, rhythm intensity-melodies and horizontal intervals, pitch range-chords and vertical intervals, rhythm danceability, rhythm intensity-rhythm, instrument-instrumentation, texture. Additionally, we incorporate fundamental music attributes such as tempo, time signature, and sequence length. Acknowledging the significance of subjective factors in music generation, we include artist, genre, and emotion—common elements in music retrieval projects [12] [13]. Furthermore, in contrast to prior conditioned music generation works (Hung et al., 2021; Ferreira et al., 2022; Bao & Sun, 2022; Mao et al., 2018; Wang et al., 2022; Choi et al., 2020; Shih et al., 2022; Yu et al., 2022; Zhang et al., 2022; Ens & Pasquier, 2020; Di et al., 2021; Wang et al., 2020b; Wu et al., 2022; Tan & Herremans, 2020; von Rütte et al., 2023; Wu & Yang, 2021) that utilize a limited set of attributes, we adopt a more extensive range to ensure comprehensive control. Based on the above reasons, we regard this set of attributes as a robust foundation, providing a suitable starting point for this work.

As the attribute set can be expanded by incorporating additional features, we encourage users to tailor it to their specific requirements. Indeed, modifying the attribute set necessitates retraining the model. To expedite verification while minimizing costs, we recommend users initially train a small model, which, despite its lower resource requirements, can still achieve commendable performance.

---

[11] https://jmir.sourceforge.net/jSymbolic.html

[12] https://mubert.com/render/themes

[13] https://open.spotify.com/search

