# OpenReview forum: "MuseCoco: Generating Symbolic Music from Text"
_ICLR.cc/2024/Conference — Submitted to ICLR 2024_

### Official Review · Reviewer_zgSb · 2023-10-24

**Soundness:** 3 good
**Presentation:** 2 fair
**Contribution:** 3 good
**Rating:** 6
**Confidence:** 4

**Summary:**

This paper proposes MuseCoco, a symbolic music generator based on text description. The text-to-music generation is decomposed into two stage: text-to-attribute stage to convert text description to music attributes, and attribute-to-music stage to generation REMI-like music sequence conditioned on the attributes. The proposed method contains large language models and trained on large symbolic datasets, showing better music generation quality and controllability compared to existing text-to-music generation methods and GPT4.

**Strengths:**

1. Originality: The proposed two-stage method to achieve text-to-music generation is simple and intuitive. With the most popular state-of-the-art language models, the study propose a convincing way to achieve high-quality and controllable symbolic music generation.
2. Quality and significance: The methodology is straightforward and sound. The experimental results is convincing.
3. Clarity: The organization of the paper is clear (though writing could be improved).

**Weaknesses:**

1. The paper writing can be improved. The passage is usually wordy and the word choice is not precise. Also, long sentences are often used which contains a lot of unnecessary logic twists. (For example, the “three advantages” part presented in the abstract section is wordy and includes long sentences.) Word spelling is another problem, for example “two-stage design”. The overall logic flow is okay to help readers capture the idea of the proposed method. A better organization at local level can help readers to catch the insight in a smoother way.
2. In our current standing, the problem of text-to-music is unsolved and one probably cannot conclude text-to-music can be perfectly decomposed into two stages with “attributes” as the intermediate representation. This is because sometimes word meanings are vague and the implication to attributes might not be this direct. The proposed method does not solve this problem scientifically but proposes a practical solution for this. To this end, the paper possibly overclaim the contribution in the understanding/modeling of the text-to-music problem.
3. Based on the previous point, the comparison with other end-to-end text-to-music models can be questionable. The two-stage model is likely to be superior in the cases where a given text prompt indicates clear attribute assignment. But what about the cases where the text prompt is abstract? The paper will be stronger if some demo of abstract text prompt is presented.

**Questions:**

Q1: in section 3.1, does position encoding not included for the m classes? Under the bi-directional BERT design, the model cannot differentiate the m classes if no position encoding is provided. For example, [cls1][cls2][x1]…[xn] and [cls2][cls1][x1]…[xn] yield the same output.

The following questions are related to the weakness discussed above:

Q2: In attribute-to-music generation, some attributes are easier music concepts while others are more complicated. Do you observe some attribute is hard or almost impossible to control? How to evaluate them? Should they be included in the attribute set?

Q3: When the text prompt is vague, how does the model behave?

---

> ### Author Response · Authors · 2023-11-22
>
> We sincerely appreciate your valuable comments. In the following response to all the concerns and questions you have posted, we use W for weakness, Q for question, and L for limitation.
>
> **W1. The paper writing can be improved.**
>
> Thank you for your guidance. We appreciate your input and we enhance the wording for the abstract part in the updated version for conciseness. Regarding the term "two-stage design," we have confirmed its grammatical correctness through dictionary research. However, we are open to specific suggestions on how to improve its usage. Your additional insights would be valuable. Thank you.
>
> **Q1. In section 3.1, does position encoding not included for the m classes?**
>
> In BERT pre-training, the first position is the first token of the input text rather than the $[CLS]$prepended to the input text. Therefore, to unleash the power of BERT by aligning with pre-training, we also assigned the first text token as the first position and did **not** compute the position embedding of $[CLS_i]^m_{i=1}$.  We prepend $[CLS_i]^m_{i=1}$ to the text after the text is inputted to BERT, instead of concatenating them before the input action. We separately calculate the embedding of $[CLS_i]^m_{i=1}$ (the word embedding and token type embedding) and the text (the word embedding, position embedding and token type embedding), and then **concatenate** the $[CLS_i]^m_{i=1}$ embedding and the text embedding. This implementation makes sure the order of $[CLS_i]^m_{i=1}$ is unchangeable. We provide the code about this in the line 244-253 of `code_submit/1-text2attribute_model/text-attribute_understanding/bert/modeling_bert.py`.
>
> **W2&W3&Q3. What about the cases where the text prompt is abstract?**
>
> In the formulation of attribute conditions, we meticulously incorporated a spectrum of explicit and nuanced descriptors to accommodate the varied preferences of users. For proficient musicians, the inclusion of precise technical music terms optimizes efficiency, ensuring the model generates music precisely in line with their articulated preferences. Conversely, users lacking a musical background often articulate their ideas using more abstract language to convey emotions or stylistic preferences. Consequently, we employ objective attributes to streamline the music creation process for musicians, supplementing them with subjective elements like emotion, artistic style, and genres. This approach facilitates expressive music generation, catering to users with diverse backgrounds and preferences.
> As you highlighted, acknowledging the crucial role of subjective elements in music generation involves describing a more abstract representation, we incorporate artist, genre, and emotion—elements commonly featured in music retrieval projects [2,3] and previous controlled music generation endeavors that incorporated subjective attributes [4-7].
>
> However, your insight into the exploration of more obscure descriptions adds an intriguing dimension to our approach. Our framework is intentionally designed for expansibility, allowing for the integration of captivating and diverse descriptions, as per your suggestion. In response to your recommendations, we undertook the creation of an additional dataset by employing ChatGPT to refine the text-to-attribute understanding stage. During this phase, we directed ChatGPT to emulate the perspective of a musician, randomly selecting attributes and their corresponding values (which could also be user-provided). Following this, we tasked ChatGPT, now assuming the role of a user without a musical background, to utilize these chosen values in crafting descriptions enriched with more abstract language. To ensure coherence, we instructed ChatGPT to provide an explanation of how each description aligns with the selected values. This augmentation and examples are presented comprehensively in Appendix D.
>
> [1] https://jmir.sourceforge.net/jSymbolic.html
>
> [2] https://mubert.com/render/themes
>
> [3] https://open.spotify.com/search
>
> [4] Hung, Hsiao-Tzu, et al. "Emopia: A multi-modal pop piano dataset for emotion recognition and emotion-based music generation." arXiv preprint arXiv:2108.01374 (2021).
>
> [5] L. N. Ferreira, L. Mou, J. Whitehead, and L. H. Lelis, “Controlling perceived emotion in symbolic music generation with monte carlo tree search,” in Proceedings of the AAAI Conference on Artificial Intelligence and Interactive Digital Entertainment, vol. 18, no. 1, 2022, pp. 163– 371 170.
>
> [6] C. Bao and Q. Sun, “Generating music with emotions,” IEEE Transactions on Multimedia, 2022.
>
> [7] H. H. Mao, T. Shin, and G. Cottrell, “Deepj: Style-specific music generation,” in 2018 IEEE 12th International Conference on Semantic Computing (ICSC). IEEE, 2018, pp. 377–382.

---

> > ### Comment · Reviewer_zgSb · 2023-11-22
> > **Response to the authors**
> >
> > Thank you for your reply.
> >
> > W1
> > In terms of paper writing, I've seen it improved in this new version. In general, I am not pointing at some specific point. What I suggest is that the authors can pay some more attention to writing so that this high-quality work can be presented in a clearer way. "Two-stage design" is of course correct. What I refer to is the "two stage design" you wrote in the beginning of the fifth paragraph in the Introduction section, where you probably missed a hyphen. There are a lot of similar places that the writing could be improved. For example, the sentence in the beginning of section 3:
> > > To achieve text-to-music generation, MuseCoco incorporates natural language and symbolic music into a two-stage framework that separates text-to-attribute understanding and attribute-to-music generation, which are trained independently.
> >
> > In this sentence there are two clauses. What does "which" in the second clause refer to? Is it (A) "text-to-attribute understanding and attribute-to-music generation" or (B) "a two-stage framework"? If it refers to (A), then the second clause is the clause of the first clause, making the whole sentence too long. Also, "t2a understanding" and "a2m generation" are more like topics rather than modules, so the meaning of the sentence is not precise. On the other hand, if it refers to (B), than this sentence should be "which is". But "which" is too far away to (B), so it's not well-written either. In both cases, the sentence is ambiguous and the writing could be improved. Hope this example is helpful.
> >
> > Q1
> > I did not fully understand, but I think what you use to differentiate the prefix tokens is either "token type embedding" or "word embedding". I hope this detail can be added to the paper in a clear way or delete the sentence regarding positional encoding. (Only showing the sentence about positional encoding in this context is a little confusing.)
> >
> > W2&W3&Q3
> > Thank you for taking my suggestion into consideration and writing a new section. This section explains to some extent the relation between obscure description and attribute assignment. Besides that, what I hope to see is to test your model on obscure descriptions. I understand this is a hard problem and if the model fails it won't affect the contribution already made.

---

> ### Author Response · Authors · 2023-11-23
>
> **W1.**
>
> We greatly value your guidance on the writing aspect, and we have incorporated your suggestions into the revised version. Continuous enhancements will be made in subsequent versions.
>
> **Q1.**
>
> As the "positional encoding" aspect is not the central focus of our work and was not introduced by us, implementation details can be found in the released code. We appreciate your suggestion, and, to avoid confusion, we have followed your advice by removing it. Thank you for your valuable input.
>
> **W2&W3&Q3.**
>
> Addressing the challenges posed by the abstract text prompt proves intricate due to the time-intensive processes involved in sample generation and questionnaire distribution, particularly for prompts requiring a crucial listening test. We may not be able to present a robust result at this point. Despite these constraints, we've included generated samples in the supplementary material (./SupplementaryMaterial/abstract_samples) for your reference. In internal testing, the majority of our team found the samples well-controlled with good musicality. However, team consensus varied, emphasizing the inherent subjectivity of such abstract text prompts. We eagerly await your feedback.
>
> It's crucial to emphasize our primary contribution in tackling the pervasive low-resource challenge in the text-to-music generation task. Additionally, we aim to provide a valuable tool for musicians to enhance workflow efficiency. The experimental results and musician feedback in Section 4 affirm the method's beneficial impact. Acknowledging the inherent difficulty in addressing all challenges comprehensively or achieving perfection, even esteemed works [1,2] in audio generation may not consistently align with input texts.
>
> Your suggestion to explore more obscure text descriptions in the future is notable. While extending our framework to diverse text descriptions, we recognize the importance of prompt engineering in constructing paired text-to-attribute data. Appendix D will incorporate this insight to guide further investigation. Once again, we appreciate your invaluable advice.
>
> [1] Agostinelli, Andrea, et al. "Musiclm: Generating music from text." arXiv preprint arXiv:2301.11325 (2023).
>
> [2] Huang, Qingqing, et al. "Noise2music: Text-conditioned music generation with diffusion models." arXiv preprint arXiv:2302.03917 (2023).

---

### Official Review · Reviewer_R4MX · 2023-10-29

**Soundness:** 3 good
**Presentation:** 2 fair
**Contribution:** 2 fair
**Rating:** 5
**Confidence:** 4

**Summary:**

The paper presents MuseCoCo, a method for performing text-to-music generation in the symbolic (rather than audio) domain. To circumvent the lack of paired text-symbolic music data, the authors utilize a two-stage model that extracts musical attributes (which can be extracted from symbolic music directly) from synthetically generated text, and then use such attributes to condition the music generation model.

**Strengths:**

- The idea of getting around the data sparsity issue through using deterministically extractable or available features is solid and is one of the main strengths of the paper.
- The overall scale of the data makes it competitive over many existing symbolic music generation systems, which are (to my knowledge) rarely trained on anything close to 1M good quality samples.

**Weaknesses:**

Overall, I am concerned about the framing of the paper, in that it avoids direct comparison with a large body of generative music research and has key methodological flaws that weaken the argument. Namely:
- While text-to-music generation in the symbolic domain is correctly assessed as a relatively young field of inquiry, by breaking down the task into text-to-attribute and attribute-to-music generation, the authors omit a large body of work in the generative music space that has dealt with attribut-conditioned generation, albeit in a more limited capacity. Of relevance:
    - [1] Rhythmic intensity, polyphonic density
    - [2] Harmonic information, melodic information
    - [3] Instrument information
    - [4] Harmonic information, tempo

- While the proposed system encapsulates multiple control signals that the aforementioned works only are able to use a subset of, the fact that none of these systems were used for even targeted comparisons over the control methods they share calls into question the novelty of the present work. It would be useful to contrast how the present work is sufficiently different from these previous papers, and to show whether the proposed method can beat prior works on the attributes they can condition on.
- The text-to-attribute results seem inflated, as the data generation technique seems to *always place the ground truth attribute value in the description.* While this is a valid type of description, it is hard to believe that human descriptions, conditional on wanting to address specific attributes, would always directly express that in natural language as explicitly as is done in the paper. I think an extension that could significantly improve the paper is to perform the ChatGPT-based data augmentation on the already-filled-in captions themselves and allow ChatGPT to obfuscate the ground-truth attributes, as the task then actually becomes text-to-attribute understanding (rather than just key word extraction).
- I have serious concerns about the the evaluation section of the work. Namely, ASA and AvgAttrCtrlAcc are metrics I have never seen before in prior work on controllable music generation, which makes it exceedingly hard to compare this work to prior works. Additionally:
    - It is unclear how ASA is calculated (from within the paper and as well from the Appendix). If ASA is calculated based on the *text-to-attribute* model, then how does that actually evaluate the music generation itself? Conversely, if ASA is calculated based on the *attribute-to-music* model, how is each attribute value being calculated deterministically from the output music?
    - It is never defined what AvgAttrCtrlAcc is in the main paper, how it is calculated, or how it is any different from ASA.
    - In Appendix B.2, it is mentioned that ASA values are calculated over different sample sizes and batch sizes for each method. While the costs for the GPT-4 baseline are understandable, I do not understand why the BART-base baseline could not have the exact same sample size as the proposed method.
- It is unclear whether the BART-base model was retrained on the same data or used directly from pretrained checkpoints (though I think it is implied the latter). In this case, without training on the same data it is hard to tell whether any of the differences are due to model design or just differences in data quality / size, aside from the fact that BART-base has 50% of the number of parameters that the proposed model does.

**Questions:**

Most questions are addressed in the above section. Namely, I think the following changes would significantly improve the paper (and would adjust my score accordingly):
- Improving the data processing techniques for the text-to-attribute model to make the problem non-extractive
- Clarifying in the work the chosen evaluation methods much more in detail, as well as including more standard evaluation methods from the literature (such as ones used in [2, 3])
- Performing any possible detailed comparison with existing models that perform limited attribute-conditioned generation, and/or scaling up the BART-base baseline to the same relative number of parameters and same data for a fairer comparison.

[1] Wu, S. L., & Yang, Y. H. (2021). MuseMorphose: Full-Song and Fine-Grained Piano Music Style Transfer with One Transformer VAE.

[2] Wu, S. L., & Yang, Y. H. (2023). Compose & Embellish: Well-structured piano performance generation via a two-stage approach.

[3] Dong, H. W., Chen, K., Dubnov, S., McAuley, J., & Berg-Kirkpatrick, T. (2022). Multitrack Music Transformer.

[4] Hsiao, W.-Y., Liu, J.-Y., Yeh, Y.-C., & Yang, Y.-H. (2021). Compound Word Transformer: Learning to Compose Full-Song Music over Dynamic Directed Hypergraphs.

---

> ### Author Response · Authors · 2023-11-22
>
> We sincerely appreciate your valuable comments. In the following response to all the concerns and questions you have posted, we use W for weakness, Q for question, and L for limitation.
>
> **W1 & W2. The authors omit a large body of work in the generative music space that has dealt with attribute-conditioned generation, albeit in a more limited capacity.  It would be useful to contrast how the present work is sufficiently different from these previous papers, and to show whether the proposed method can beat prior works on the attributes they can condition on.**
>
> Thank you for your valuable suggestions. Firstly, MuseCoco significantly focuses on overcoming challenges in generating symbolic music from text descriptions. Consequently, a thorough comparison with methods specifically designed for this scenario is essential, and we have already undertaken this in Table 4.
>
> Secondly, as attribute-to-music generation constitutes a crucial module in our system, it becomes imperative to conduct an ablation study to validate its effectiveness, as depicted in Table 5. In our analysis, we compare our control method labeled as "Prefix Token" with other widely used methods such as "Embedding" and "Layer Norm." The results reveal that in a multi-attribute control scenario, the "Prefix Token" method demonstrates significantly superior controllability, manifesting at least a 20% improvement.
>
> The paper you referenced is indeed valuable for a thorough comparison to further validate our method. However, it's crucial to note that it falls within the category of methods we are already comparing. For instance, in [1], the conversion of segment-level conditions into an embedding vector, followed by concatenation with token embeddings at each time step, aligns with what we denoted as the "Embedding" method in Table 5.
>
> Additionally, three other Works[2-4], namely "Compose & Embellish," "Multitrack Music Transformer," and "Compound Word Transformer," share similarities with the "Prefix Tokens" method employed in our approach. This similarity lies in the explicit representation of condition tokens in the input sequence instead of condensing them into a vector. Notably, this architecture is widely employed for decoder-only generation tasks, akin to Prefix language modeling [5].
>
> We appreciate your observation regarding missing references and will promptly address this in Section 3.2 and Section 4.3.2 in the updated version. Thank you for bringing it to our attention.
>
> [1] Wu, S. L., & Yang, Y. H. (2021). MuseMorphose: Full-Song and Fine-Grained Piano Music Style Transfer with One Transformer VAE.
>
> [2] Wu, S. L., & Yang, Y. H. (2023). Compose & Embellish: Well-structured piano performance generation via a two-stage approach.
>
> [3] Dong, H. W., Chen, K., Dubnov, S., McAuley, J., & Berg-Kirkpatrick, T. (2022). Multitrack Music Transformer.
>
> [4] Hsiao, W.-Y., Liu, J.-Y., Yeh, Y.-C., & Yang, Y.-H. (2021). Compound Word Transformer: Learning to Compose Full-Song Music over Dynamic Directed Hypergraphs.
>
> [5] Wang, Thomas, et al. "What language model architecture and pretraining objective works best for zero-shot generalization?." International Conference on Machine Learning. PMLR, 2022.

---

> > ### Author Response · Authors · 2023-11-22
> >
> > **W3.  I think an extension that could significantly improve the paper is to perform the ChatGPT-based data augmentation on the already-filled-in captions themselves and allow ChatGPT to obfuscate the ground-truth attributes, as the task then actually becomes text-to-attribute understanding (rather than just key word extraction).**
> >
> > Thanks for pointing out this interesting topic that we can discuss.
> >
> > Initially, in the process of selecting attributes as conditions, we carefully considered the inclusion of explicit and obscure descriptions to cater to diverse user preferences. For professional musicians, efficiency is enhanced by providing technical music terms, enabling the model to generate music explicitly aligned with their descriptions. Conversely, users without a music background express their ideas using more obscure language to convey emotions or preferred styles. Consequently, we utilize objective attributes to streamline the music creation process for musicians and introduce subjective attributes such as emotion, artistic style, and genres to facilitate expressive music generation for users of varying backgrounds and preferences.
> >
> > Furthermore, our framework is designed to be expansible, allowing for the inclusion of more intriguing and diverse descriptions, as you suggested. In response to your recommendations, we endeavored to construct an additional dataset by leveraging ChatGPT to fine-tune the text-to-attribute understanding stage. In this phase, we instructed ChatGPT to simulate the role of a musician, randomly selecting attributes and their corresponding values (which can also be user-provided). Subsequently, we tasked ChatGPT, assuming the role of a user without a music background, to employ these chosen values in generating descriptions enriched with more abstract language. To ensure logical alignment, we let it provide an explanation of how the description corresponds to the selected values. This augmentation is presented in Appendix D in the updated version.

---

> ### Author Response · Authors · 2023-11-22
>
> **W4. (1) ASA and AvgAttrCtrlAcc are metrics I have never seen before in prior work on controllable music generation, which makes it exceedingly hard to compare this work to prior works.**
>
> The metrics ASA (Attribute Specific Accuracy) and AvgAttrCtrlAcc (Average Attribute Control Accuracy) have been specifically introduced to gauge the control performance of the proposed system. Given the objective of guiding the music generation process through attributes specified in textual descriptions, it is imperative to assess whether the generated samples adhere to the attribute values outlined in the text.
>
> ASA quantifies the correctness of predicted attributes within each sample and subsequently averages the accuracy across the entire test set. This metric is proposed for this text-to-symbolic music generation framework, which provides insight into the alignment between the generated sample and the textual descriptions, ensuring that the generated content aligns appropriately with the specified attributes.
> On the other hand, AvgAttrCtrlAcc measures the accuracy of each controlled attribute individually for the attribute-to-music generation stage. By evaluating the precision of attribute control for each specific parameter, this metric offers a nuanced understanding of the system's ability to accurately manipulate individual attributes.
>
> In contrast to earlier methodologies [1,2] that employ end-to-end approaches for synthesizing symbolic music from textual descriptions, a significant challenge arises in assessing the alignment between the generated music and the specified attributes in the text. This difficulty arises because the prior methods do not explicitly identify these attributes. In contrast, MuseCoco takes a distinctive approach by explicitly defining objective attributes within the text descriptions. These defined attributes can be easily extracted from the generated music, facilitating a straightforward evaluation of alignment through direct accuracy calculations. This departure from traditional end-to-end methods not only enables a more precise assessment but also establishes a clear and robust relationship between textual descriptions and the resulting musical output.
>
>
> [1] Shangda Wu and Maosong Sun. Exploring the efficacy of pre-trained checkpoints in text-to-music generation task. arXiv preprint arXiv:2211.11216, 2022.
>
> [2] Yixiao Zhang, Ziyu Wang, Dingsu Wang, and Gus Xia. BUTTER: A representation learning framework for bi-directional music-sentence retrieval and generation. In Proceedings of the 1st Workshop on NLP for Music and Audio (NLP4MusA), pp. 54–58. Association for Computational Linguistics, 16 October 2020.
>
> **(2) How ASA is calculated.**
>
> ASA plays a crucial role in the Main Results section, specifically for comparing text-to-music generation against various baselines. Specifically, we produce samples for distinct baselines using identical text descriptions as inputs. These text descriptions are synthesized following the methods outlined in Section 3.3. Given that the text descriptions are generated based on the attributes defined in Table 8, we can easily assess the alignment of the generated music with the text descriptions. This evaluation involves extracting objective attributes from the generated music and comparing them with the attributes outlined in the text descriptions. For subjective attributes, we conducted a listening test to gauge control accuracy. Subsequently, we determine the proportion of accurately predicted attributes in each sample and compute the average accuracy across the entire test set.
>
> To facilitate understanding, consider the following example:
>
> “Text descriptions: 3/4 is the time signature of the music. This song is unmistakably classical in style. This music is low-tempo. The use of piano is vital to the music.”
>
> Within these descriptions, three attributes—time signature, tempo, and instrumentation—are objective, while one, genre, is subjective. Objective attribute values are extracted from generated samples, and a listening test is conducted to categorize the subjective attribute (genre). Assume that correct predictions are made for time signature, tempo, and genre, while instrumentation predictions fall short. The resulting sample-wise accuracy is 75%. By aggregating these accuracies across the entire testing set, the averaged sample-wise accuracy (ASA) is determined.

---

> ### Author Response · Authors · 2023-11-22
>
> **(3) It is never defined what AvgAttrCtrlAcc is in the main paper, how it is calculated, or how it is any different from ASA.**
>
> AvgAttrCtrlAcc serves as a key metric for evaluating control accuracy during the attribute-to-music generation stage. It specifically gauges the accuracy of predicting each attribute by calculating the proportion of correctly predicted instances across all testing samples. The resulting average accuracy for each specific attribute is presented in Table 11. It's important to distinguish AvgAttrCtrlAcc from ASA in two fundamental aspects: 1) AvgAttrCtrlAcc is computed by averaging accuracy over a particular attribute across the entire testing set, whereas ASA is an average calculated over samples containing multiple attributes within the testing set. 2) AvgAttrCtrlAcc is designed to evaluate the attribute-controlling performance during the attribute-to-music generation stage, focusing on the precision of individual attributes. In contrast, ASA is formulated to assess the overall alignment between generated samples and provided text descriptions.
>
> Utilizing abbreviations may lead to a misperception that we are introducing an entirely novel metric. In reality, both metrics fundamentally measure accuracy in distinct aspects, which is not groundbreaking. To prevent any misunderstanding, we have decided to rename them as **"Text-controlled Accuracy"** and **"Attribute-controlled Accuracy"** in the updated version. Thanks for your advice.
>
>
> **(4) In Appendix B.2, it is mentioned that ASA values are calculated over different sample sizes and batch sizes for each method. While the costs for the GPT-4 baseline are understandable, I do not understand why the BART-base baseline could not have the exact same sample size as the proposed method.**
>
> Thank you for bringing this to our attention. The variance in sample size between MuseCoco and the BART-based baseline results from our listening test methodology. In the listening test, we typically sample three generated responses for each prompt. To manage time constraints, we opted to generate only five clips per prompt for the BART-based baseline. Your valuable feedback prompted us to reconsider, and in light of your input, we have increased the sample size for the BART-based baseline to 10 samples per prompt, aligning it with MuseCoco's sample size.
>
> Consequently, we recalculated the ASA values, and the updated result for the BART-based baseline is now 32.47%, compared to the previous value of 31.98%. We will reflect this correction in the main paper. Your timely and thoughtful feedback has contributed significantly to the accuracy and completeness of our study.

---

> > ### Author Response · Authors · 2023-11-22
> >
> > **W5. It is unclear whether the BART-base model was retrained on the same data or used directly from pretrained checkpoints (though I think it is implied the latter). In this case, without training on the same data it is hard to tell whether any of the differences are due to model design or just differences in data quality / size, aside from the fact that BART-base has 50% of the number of parameters that the proposed model does.**
> >
> > Achieving a truly equitable comparison proves to be a formidable task, particularly when confronted with the challenge of aligning the model size of GPT-4 while ensuring parity with a BART-based model.
> >
> > Several notable works deviate from the conventional approach of comparing baselines solely based on identical training data sizes. For instance, in the case of MusicLM [2] and Noise2Music [3], comparisons were conducted by directly querying APIs or running inference steps using provided checkpoints, without strictly adhering to matching data sizes.
> >
> > In our pursuit of a fair assessment, we advocate for evaluating models at their pinnacle performance levels, enabling the results to stand on their own merit. The BART-based model, chosen for its exceptional performance among various pretrained checkpoints as detailed in [1], is employed directly from pretrained checkpoints. Despite this, our system distinguishes itself by leveraging a substantial volume of unlabeled data within the proposed framework, thereby showcasing optimal performance. Renowned for its formidable language modeling capabilities, GPT-4 owes its prowess to an extensive dataset and robust model parameters.
> >
> > The results unequivocally demonstrate MuseCoco's superiority over the two mentioned baselines in terms of controllability and musicality. It's noteworthy that the version of MuseCoco used for this comparison, as indicated in Table 6, does not represent its maximum potential with a larger model size. Nevertheless, even in this configuration, MuseCoco outperforms the baselines, attesting to its impressive capabilities.
> >
> > [1] Shangda Wu and Maosong Sun. Exploring the efficacy of pre-trained checkpoints in text-to-music generation task. arXiv preprint arXiv:2211.11216, 2022.
> >
> > [2] Agostinelli, Andrea, et al. "Musiclm: Generating music from text." arXiv preprint arXiv:2301.11325 (2023).
> >
> > [3] Huang, Qingqing, et al. "Noise2music: Text-conditioned music generation with diffusion models." arXiv preprint arXiv:2302.03917 (2023).

---

> > > ### Comment · Reviewer_R4MX · 2023-11-23
> > >
> > > The reviewer sincerely thanks the authors for their detailed response to the concerns brought up, and especially in clearing up a lot of the confusion I had with the original draft of the paper. While I still have some concerns with regards to the overall framing of the paper given the lack of evaluations with abstract text prompts (as I don't think it is necessarily correct that musicians would always use "objective" specific terms while non-musicians would use "subjective terms, e.g. both musicians and non-musicians could say "in 3" or "in waltz feel" to both describe a 3/4 time signature for example), I have increased my score in light of the authors response to my other questions.

---

> ### Author Response · Authors · 2023-11-23
>
> Thank you for acknowledging our response.
>
> Addressing the issues raised in the abstract text prompt presents challenges in providing concrete experimental results due to the time-intensive nature of sample generation and questionnaire distribution, particularly for prompts of this nature where a listening test is crucial. We may not be able to present a robust result at this point. Despite these constraints, we have included some generated samples in the supplementary material (./SupplementaryMaterial/abstract_samples) for your reference. During internal testing, the majority of our team members found that the samples were well-controlled and exhibited good musicality. However, consensus among the team varied, highlighting the inherent subjectivity associated with abstract text prompts of this nature. Your feedback on these samples is eagerly awaited.
>
> It is important to underscore that our primary contribution lies in tackling the prevalent low-resource challenge in the text-to-music generation task. Additionally, we aim to offer a valuable tool for musicians to enhance their workflow efficiency. The experimental results and feedback from musicians in Section 4 demonstrate that our method has proven beneficial to some extent. It is essential to acknowledge the inherent difficulty in addressing all challenges comprehensively or achieving perfection in a single endeavor. Even notable works [1,2] in generating audio from text descriptions may not consistently align the generated music with the input texts.
>
> Your suggestion regarding the consideration of more obscure text descriptions as a future avenue of exploration is noteworthy. In our efforts to extend our framework to encompass a broader range of text descriptions, we discovered the significance of prompt engineering in constructing paired text-to-attribute data. Experimenting with more meticulously designed prompts is imperative, and we will incorporate this insight into Appendix D to guide further investigation. Once again, we express our gratitude for your valuable advice.
>
> [1] Agostinelli, Andrea, et al. "Musiclm: Generating music from text." arXiv preprint arXiv:2301.11325 (2023).
> [2] Huang, Qingqing, et al. "Noise2music: Text-conditioned music generation with diffusion models." arXiv preprint arXiv:2302.03917 (2023).

---

### Official Review · Reviewer_WPYi · 2023-10-30

**Soundness:** 2 fair
**Presentation:** 3 good
**Contribution:** 2 fair
**Rating:** 5
**Confidence:** 3

**Summary:**

The authors present two models which are used in tandem to enable symbolic music generation (ABC) via text prompt. The first model converts the text prompt to a musical attribute space. The second model generates the symbolic music from the musical attributes. The benefit of this two stage approach is that we can alleviate the lack of text to music data.

**Strengths:**

* Good approach to alleviate data deficit of text description to abc music data by leveraging existing models for extracting features from music, and expanding those features to text though templates and generative models for text

**Weaknesses:**

* Evaluation details could be clearer - though the number of participants is stated in the appendix (19) the method by which they were sampled is not
* Claims that the model "closely resembles human performances in musicality" are not substantiated, since no comparison with human performance was done.
* Training was performed on private datasets so the result is not reproducible - it would be excellent if a comparison with publicly available data could be done.
* No details were given about how musical attributes were automatically extracted from the abc music (other than those which are trivially in abc such as the time signature and key)
* There was no commentary on how this set of attributes were arrived at - although it is mentioned that this list could be expanded easily, it would require a complete re-train of the model. Is there any justification that this is a good base set of attributes?

**Questions:**

* Is there a reason that the Lakh dataset was not used too for training (I know you use the MMD but this is not a superset)?
* How would you expand this model for multi-instrument music?

---

> ### Author Response · Authors · 2023-11-20
>
> We sincerely appreciate your valuable comments. In the following response to all the concerns and questions you have posted, we use W for weakness, Q for question, and L for limitation.
>
> **W1. Evaluation details could be clearer - though the number of participants is stated in the appendix (19) the method by which they were sampled is not.**
>
> Thanks for asking. As we stated in Appendix B.1, “To ensure the reliability of the assessment, only individuals with at least music profession level 3 were selected, resulting in a total of 19 participants”. Since all the participants are above level 3, 19 is the exact number of participants without selecting by designed sample method.
>
> **W2. Claims that the model "closely resembles human performances in musicality" are not substantiated, since no comparison with human performance was done.**
>
> Thank you for bringing this out. Comparing the cost-effectiveness of our results with human compositions, where employing musicians to compose based on provided text descriptions incurs significant expenses, we have opted for a subjective listening test. This test is designed to evaluate musicality by gauging the similarity between the generated samples and human compositions.
> To measure this similarity, we have designed a scoring system ranging from 1 to 5 as illustrated in Section 4.1, with higher scores indicating a closer resemblance to human compositions, which is explicitly stated in questionnaire. To ensure the professionalism of the feedback, we only consider responses from participants with a proficiency level of at least 3, as defined in Table 9. This ensures that participants can provide expert evaluations on the gap between human compositions and the generated samples.
> MuseCoco attains a score of 4.06, significantly closer to 5 than the baselines. This scoring approach aligns with the methodology employed in numerous prior studies [1-4].
>
>
> [1] Dong, Hao-Wen, et al. "Musegan: Multi-track sequential generative adversarial networks for symbolic music generation and accompaniment." Proceedings of the AAAI Conference on Artificial Intelligence. Vol. 32. No. 1. 2018.
>
> [2] Yu, Botao, et al. "Museformer: Transformer with fine-and coarse-grained attention for music generation." Advances in Neural Information Processing Systems 35 (2022): 1376-1388.
>
> [3] Schneider, Flavio, Zhijing Jin, and Bernhard Schölkopf. "Mo\^ usai: Text-to-Music Generation with Long-Context Latent Diffusion." arXiv preprint arXiv:2301.11757 (2023).
>
> [4] Shih, Yi-Jen, et al. "Theme transformer: Symbolic music generation with theme-conditioned transformer." IEEE Transactions on Multimedia (2022).
>
> **W3. Training was performed on private datasets so the result is not reproducible - it would be excellent if a comparison with publicly available data could be done.**
>
> We are committed to ensuring reproducibility, and therefore, we provide codes, and will release trained checkpoints later in the future, for transparency in our results. This will not only facilitate the replication of our findings but also enable the re-training of the model using publicly available data. While the released code supports re-training, it's worth noting that due to the limited availability of public data, utilizing the checkpoints directly is recommended for more robust reproducibility.
>
> It's important to highlight that our past experiments indicate that larger datasets contribute to enhanced performance. To illustrate, we present the comparative results below:
>
> |                                 | Musicality    | Controllability | Overall       | ASA(%) |
> |---------------------------------|---------------|-----------------|---------------|--------|
> | MuseCoco (6 layers, 20w data)   | 3.55$\pm$0.81 | 3.55$\pm$1.11   | 3.56$\pm$0.80 | 77.45  |
> | MuseCoco (16 layers, 100w data) | 4.06$\pm$0.82 | 4.15$\pm$0.78   | 4.13$\pm$0.75 | 77.59  |
> |                                 |               |                 |               |        |
>
> In examining this table, it becomes evident that as both model size and data size increase, there is a corresponding improvement in musicality and control accuracy. Consequently, leveraging pretrained checkpoints becomes imperative to attain a relatively superior result.

---

> > ### Author Response · Authors · 2023-11-20
> >
> > **W4. There was no commentary on how this set of attributes were arrived at - although it is mentioned that this list could be expanded easily, it would require a complete re-train of the model. Is there any justification that this is a good base set of attributes?**
> >
> > Indeed, modifying the attribute set necessitates retraining the model. To expedite verification while minimizing costs, we recommend users initially train a small model, which, despite its lower resource requirements, can still achieve commendable performance.
> >
> > It is customary to retrain models for specific scenario-based tasks, especially when altering certain factors. For instance, in the best paper at ACMM 2021 [1], the model needs to be retrained as well when attributes such as music density and motion speed in videos are modified for background music generation. In this type of work [5-13], where attributes are paired with music sequences in the training data, any change in attributes results in a shift in data distribution. Consequently, the model necessitates retraining to update its parameters and effectively adapt to the altered input-output relationships.
> >
> > Our chosen attributes are grounded in musical theory and draw inspiration from prior research. Leveraging JSymbolic [2], a widely-used tool for extracting statistical information from music data in the field of Music Information Retrieval (MIR), we ensure that each category we utilize features at least one attribute. The representation follows the format "{attribute}-{category}" to signify the coverage within each category. Examples include {pitch range, key}-{pitch statistics}, {rhythm intensity}-{melodies and horizontal intervals}, {pitch range}-{chords and vertical intervals}, {rhythm danceability, rhythm intensity}-{rhythm}, {instrument}-{instrumentation, texture}. Additionally, we incorporate fundamental music attributes such as tempo, time signature, and sequence length. Acknowledging the significance of subjective factors in music generation, we include artist, genre, and emotion—common elements in music retrieval projects [3,4].
> >
> > Furthermore, in contrast to prior conditioned music generation works [5-13] that utilize a limited set of attributes, we adopt a more extensive range to ensure comprehensive control. Based on the above reasons, we regard this set of attributes as a robust foundation, providing a suitable starting point for this work.
> >
> > [1] Di, Shangzhe, et al. "Video background music generation with controllable music transformer." Proceedings of the 29th ACM International Conference on Multimedia. 2021.
> >
> > [2] https://jmir.sourceforge.net/jSymbolic.html
> >
> > [3] https://mubert.com/render/themes
> >
> > [4] https://open.spotify.com/search
> >
> > [5] Hung, Hsiao-Tzu, et al. "Emopia: A multi-modal pop piano dataset for emotion recognition and emotion-based music generation." arXiv preprint arXiv:2108.01374 (2021).
> >
> > [6] L. N. Ferreira, L. Mou, J. Whitehead, and L. H. Lelis, “Controlling perceived emotion in symbolic music generation with monte carlo tree search,” in Proceedings of the AAAI Conference on Artificial Intelligence and Interactive Digital Entertainment, vol. 18, no. 1, 2022, pp. 163– 371 170.
> >
> > [7] C. Bao and Q. Sun, “Generating music with emotions,” IEEE Transactions on Multimedia, 2022.
> >
> > [8] H. H. Mao, T. Shin, and G. Cottrell, “Deepj: Style-specific music generation,” in 2018 IEEE 12th International Conference on Semantic Computing (ICSC). IEEE, 2018, pp. 377–382.
> >
> > [9] W. Wang, X. Li, C. Jin, D. Lu, Q. Zhou, and Y. Tie, “Cps: Full-song and style-conditioned music generation with linear transformer,” in 2022 IEEE International Conference on Multimedia and Expo Workshops (ICMEW). IEEE, 2022, pp. 1–6.
> >
> > [10] K. Choi, C. Hawthorne, I. Simon, M. Dinculescu, and J. Engel, “Encoding musical style with transformer autoencoders,” in International Conference on Machine Learning. PMLR, 2020, pp. 1899–1908.
> >
> > [11] J. Ens and P. Pasquier, “Mmm: Exploring conditional multi-track music generation with the transformer,” arXiv preprint arXiv:2008.06048, 2020.
> >
> > [12] H. H. Tan and D. Herremans, “Music fadernets: Controllable music generation based on high-level features via low-level feature modelling,” arXiv preprint arXiv:2007.15474, 2020.
> >
> > [13] D. von Rütte, L. Biggio, Y. Kilcher, and T. Hofmann, “Figaro: Controllable music generation using learned and expert features,” in The Eleventh International Conference on Learning Representations, 2023.

---

> > > ### Author Response · Authors · 2023-11-20
> > >
> > > **W5. No details were given about how musical attributes were automatically extracted from the abc music (other than those which are trivially in abc such as the time signature and key).**
> > >
> > > We transform the ABC notation music generated by the baselines into MIDI music using the music21 library [1]. After getting the midi files, the same procedure for the extraction of attributes follows the rules outlined in Table 8. Elements such as instrument, bar, time signature, and tempo are directly derived from MIDI sequences. The pitch range is computed by subtracting the minimum pitch from the maximum pitch within a music sequence. Rhythm danceability is determined by calculating the ratio of downbeats, while rhythm density is measured as the average note density. Key estimation is based on note distribution using musical rules. Additionally, time is derived from the time signature and the total number of bars. We will add the detail to Appendix C in the updated version, thanks for your question.
> > >
> > > [1] http://web.mit.edu/music21/
> > >
> > > **Q1. Is there a reason that the Lakh dataset was not used too for training (I know you use the MMD but this is not a superset)?**
> > >
> > > Actually, Lakh dataset is included in MMD corpus when we got from the authors, so we don’t mention it additionally.
> > >
> > > **Q2. How would you expand this model for multi-instrument music?**
> > >
> > > The model inherently possesses the capability to handle multi-instrument music, as demonstrated on the demo page (e.g., Sample 3, Sample 4, and Sample 5 in Generation Diversity section). As illustrated in Table 7, users can select different instruments collectively as conditions for generating music, so that users can specify a diverse range of instrumentation through text descriptions.

---

> > > > ### Comment · Reviewer_WPYi · 2023-11-22
> > > > **Response to rebuttals**
> > > >
> > > > Thank you for your detailed responses. I'll respond to all in turn.
> > > >
> > > > W1 - you've not clarified how you found these people e.g. did they respond to a tweet? Are they from your university? etc.
> > > > W2 - your response is fair, but the wording of your question is not included within the supplementary table. Exactly what you asked the participants is an important detail to include if you are using this score as a proxy for your central claim.
> > > > W3 - I much appreciate your commitment and appreciate the model has been included, I don't think you could do more to address this weakness.
> > > > W4 - This is a very useful response and I encourage you to add these details to your paper and/or supplementary material.
> > > > W5 - As you have agreed, please add these details to the supplementary info.
> > > >
> > > > Q1 - is this generally the case or was it something the holders of the MMD dataset (reference?) did for your convenience?
> > > > Q2 - thank you for directing me to these examples, and apologies for my oversight.

---

> > > > > ### Author Response · Authors · 2023-11-23
> > > > >
> > > > > **W1. You've not clarified how you found these people.**
> > > > >
> > > > >
> > > > > These students from various universities. We crafted a comprehensive questionnaire delineating the task details and instructions for scoring, encapsulated within a document file. Accompanying this, we provided a folder containing samples to score, and an Excel file featuring sample indices and corresponding score columns, facilitating their input of evaluations. Subsequently, we gathered and analyzed the responses, employing defined metrics for calculation. We have already incorporated the questions posed in the questionnaire into the Appendix B.1.
> > > > >
> > > > > **W4 & W5**
> > > > >
> > > > > Thank you for your suggestions. We have added W4 into Appendix F and W5 to Appendix C, please refer.
> > > > >
> > > > > **Q1.**
> > > > >
> > > > > We queried the author regarding the absence of explicit information about the dataset structure in their paper. They responded, stating that the Lakh dataset should be included within the previously mentioned MMD dataset.

---

### Official Review · Reviewer_Y1vJ · 2023-11-03

**Soundness:** 4 excellent
**Presentation:** 4 excellent
**Contribution:** 4 excellent
**Rating:** 8
**Confidence:** 4

**Summary:**

A symbolic music generation technique based on language models is presented.
The method is based on two-stage processing. One is extracting musical attributes and another is attribute-music synthesis. The former is based on BERT. This part will not be that difficult. What is important is the latter part. The authors leverated vast amount of unlabeld musical data, and estimated the attributes automatically since some of attributes (e.g. tempo, key, instrument) are easily extracted from symbolic music data without specific annotation. Using these unlabeled data and these attributes extracted from them, a transformer that generates symbolic music is trained.
Because the proposed method has been properly trained with a vast amount of musical data, its output is clearly superior musically to the poor results by GPT4, which contains little musical knowledge and only feeded a few examples through prompts. It also produces results that are faithful to the instructions.

**Strengths:**

I myself have had a painful experiment to devise GPT-4 prompt to generate music that is at least listenable. (Still, it was not up to the quality of the GPT-4 samples that the author has on the github page.) The authors must have been quite ingenious. And I was honestly impressed that MuseCoCo could generate music of a quality far superior to that. The results are quite impressive indeed.

The approach appears to be solid. And the results and the quality of the paper are excellent. The explanations are clear enough (though I would like a few more examples). It would be significant in terms of accelerating the composition workflow.

**Weaknesses:**

The maximum length of music that the model can generate is limited because it is based on a Transformer.

**Questions:**

- Having had my own experience with GPT-4, I agree with the results presented by the author, but some readers may possibly criticize the results as "cherry-picking". Objective evaluations such as Table 4 and comments by professional musicians are actually making the results convincing, but I still have an impression that the paper lacks something. Perhaps because there are no examples in the body of the paper. Simply posting multiple sheets or piano rolls would probably improve the impression of the paper.
- Readers with some musicologic background knowledge will note that the example generated in Figure 4 violates prohibitions of classical counterpoint techniques. Of course, it would be possible to use such a technique in film or game music to represent "forest strolls" and "whispering winds" but this may not be a good example to convey the strength of this technique, since a piece with two voices in the fifth and sixth intervals does not appear to be very sophisticated. (In other words, this example alone gives the impression that it is not a greatly superior technique.) The key is supposed to be C major, but it looks like A minor or A Aeolian to me. This too would be disconcerting to a reader with musical knowledge. ...... And I realized after writing this that this is the baseline GPT-4 result to which you are comparing. That makes sense, but I would still like to see specific input examples and output results of the proposed method in the main body of the paper.

**Details Of Ethics Concerns:**

No information could be found on the rights to the MIDI data used for the training. It is not clear to me whether these are all rightful to use in training. The authors report that some of the training data are copyrighted.

---

> ### Author Response · Authors · 2023-11-20
>
> We sincerely appreciate your valuable comments. In the following response to all the concerns and questions you have posted, we use W for weakness, Q for question, and L for limitation.
>
> **Q1. Perhaps because there are no examples in the body of the paper. Simply posting multiple sheets or piano rolls would probably improve the impression of the paper.**
>
> Thanks for your acknowledgement. Given that the generated music typically comprises multiple tracks and spans several bars, incorporating them into the main paper poses a challenge due to space constraints. Due to the page limit, we will put a link at the end of the demo page so that users can download the sheets for each sample in the demo page for future use.
>
> **Q2. And I realized after writing this that this is the baseline GPT-4 result to which you are comparing. That makes sense, but I would still like to see specific input examples and output results of the proposed method in the main body of the paper.**
>
> Yes, it is the baseline GPT-4 result that may not be controlled well. Given the limited page length, we showcase input examples and output results of our proposed method on the demo page, prioritizing the communication of the core idea within the main paper. We plan to release checkpoints in the future so that users can directly get the lead sheets from their given descriptions. Thanks for your attention.

---

> ### Comment · Reviewer_Y1vJ · 2023-11-22
> **Response to the authors**
>
> Thank you for your response.
>
> I understand that we can download generated data from the link, but my concern is that the GitHub demo page can be modified as much as you want without anyone except the most careful ones aware of it. Since Git basically allows log modification, it can be updated after the submission deadline, during peer review, or even after publication. This is not fair. The main contribution of the research results should be included either in the main text or in an appendix (supplementary materials). Appendices in particular have virtually no page limit, and it is not uncommon in this field for papers to have appendices that sometimes run into the dozens of pages. I don't think you can make the argument that the page limit is a reason not to include specific examples.
>
> It seems very unnatural to me that the authors carefully include an example of the baseline method, but no specific information about the results obtained with the proposed method at all. A reader trained to read critically would suspect that the authors are withholding some inconvenient information.
>
> I'm not asking you to list all the results, but I think it is normal for most papers in this field to include at least a few examples of piano rolls, if not the full score.

---

> ### Author Response · Authors · 2023-11-22
>
> Thank you for your response.
>
> When you suggested including the sheet in the "main body" of the paper, we opted to focus more on the general idea and results due to page limitations. However, if it's acceptable to add this information in the Appendix, we have done so in the updated version (i.e., Appendix E). We want to clarify that our argument is not an attempt to conceal inconvenient information; otherwise, we could have omitted the sheets from the demo page.
>
> It's worth noting that not all papers of this nature necessarily include examples; it depends on the specific topics. For instance, works [1,2] also don't include examples, perhaps because it's challenging to assess effectiveness by examining only a few bars. In our case, explaining how the generated music aligns well with the text inputs is complex with just a few bars shown in the lead sheets. Some objective attributes need to be calculated over the entire sequence, and subjective attributes should be evaluated through listening experiences. We appreciate your proactive response and understanding, contributing to the enhancement and persuasiveness of our work.
>
> [1] von Rütte, Dimitri, et al. "FIGARO: Controllable Music Generation using Learned and Expert Features." The Eleventh International Conference on Learning Representations. 2022.
>
> [2] Wu, Shih-Lun, and Yi-Hsuan Yang. "MuseMorphose: Full-song and fine-grained piano music style transfer with one transformer VAE." IEEE/ACM Transactions on Audio, Speech, and Language Processing 31 (2023): 1953-1967.

---

### Meta-Review · Area_Chair_XXHN · 2023-12-08

**Metareview:**

This is an interesting paper, borderline for publication at ICLR, that I recommend to reject. The paper presents a two steps method to generate MIDI (musical notation for computers) scores from text prompts. The first step generates attributes (instruments, time signature, tempo, genre...), the second step is a language model over notes, (pre-)conditioned on those attributes (203M or 1.2B parameters depending on version). The authors contributed productively to the rebuttal and the paper was improved significantly (in clarity). However, the main contributions of the paper is in preparing a proprietary dataset and training a language model on it: the scope is rather limited. Several reviewers note those limitations, even Reviewer Y1vJ who rated the paper "8", mentions several limitations. The human evaluation contain only 19 (high skill musicians) participants, which makes the evaluation more qualitative than quantitative. In its current state, the paper doesn't teach us something consequent enough, doesn't release something significant that the community can build upon, and thus is not suitable for publication at ICLR.

**Justification For Why Not Higher Score:**

In its current state, the paper doesn't teach us something consequent enough, doesn't release something significant that the community can build upon. The evaluations are too light, the baselines somewhat trivial, the method has nothing particular.

**Justification For Why Not Lower Score:**

N/A

---

### Decision · Program_Chairs · 2024-01-16

Reject